# WHEN WE DON'T SEE THE SAME PICTURE: ALIGNING AGENTS WITH DIVERGENT VISUAL SPACES

## ABSTRACT

With the rapid rise of agentic systems, interaction and collaboration between agents has become a central challenge in multimodal large language model (MLLM) research. In this work, we study a unique form of interaction where two agents hold *divergent visual spaces*. To investigate this, we adopt image reference games as a testbed and design experiments with five distinct perceptual distortions inspired by real human visual impairments (e.g., cataract, color blindness). We evaluate in two settings, online and offline, with four post-training algorithms: SFT, DPO (offline) and KTO, GRPO (online), providing the first systematic study of alignment under divergent visual perceptions. Our results reveal that (i) offline adaptation can provide strong improvements, with DPO consistently outperforming other methods when supported by high-quality preference data; (ii) among online adaptation methods, KTO yields strongest average gains across datasets and (iii) qualitative analysis shows that adapted agents shift their descriptions toward perceptual features accessible to their conversation partners. Taken together, these findings highlight that offline methods are the preferable solution when supervision is available while online approaches serve as a complementary strategy for dynamic settings where distortions or partner characteristics are unknown in advance. We release code and preference datasets to support future research.

## 1 INTRODUCTION

Large Language Models (LLMs) and their multimodal extensions are increasingly deployed in real-world settings, powering applications in video analysis, human–AI collaboration, and autonomous decision-making (Fu et al., 2024; Liu et al., 2024b; Fu et al., 2025; Huang et al., 2025b;a; Hu et al., 2025; Bian et al., 2025a;b; Wu et al., 2025). These deployments require agents not only to process complex sensory streams but also to interact effectively with humans and other agents. A common assumption in prior work is that agents share a consistent world-view (Ju & Aral, 2025; Hsu et al., 2025; Zhang et al., 2024). Yet in practice, communication often breaks down precisely because partners perceive the world differently. For humans, such divergences arise naturally in conditions of impaired or limited vision (such as cataract or color blindness) where shared understanding depends on adapting communication to perceptual asymmetries. More importantly, there are over 1 billion (Nwabueze, 2023) people with perceptual disabilities, yet very few works have explored the alignment with this large group of people.

Motivated by these human-centered challenges, we study how artificial agents can communicate under mismatched perceptual abilities. Specifically, we explore alignment between agents that diverge in their visual perception. To ground this problem, we simulate five types of image distortions inspired by real human visual impairments, spanning both spatial and pixel-level degradations. We adopt the image reference game setup (Corona et al., 2019) with divergent visual spaces, where a *speaker* describes a target image and a *listener* must identify it among a pair. Crucially, the speaker observes the undistorted images, while the listener only sees distorted versions, creating a perceptual gap that the speaker must bridge.

Our study examines how agents can adapt to such mismatches. We consider both offline adaptation, using supervised fine-tuning (SFT) and direct preference optimization (DPO) Rafailov et al. (2023), and online adaptation, using preference-based algorithms such as Kahneman-Tversky Optimization (KTO) (Ethayarajh et al., 2024) and Group Relative Policy Optimization (GRPO) (Shao et al., 2024). To support preference learning, we construct a dataset by prompting Qwen2.5-VL (Wang et al., 2024b) with distorted inputs to generate positive examples and with undistorted inputs to generate negative examples, explicitly modeling the gap between successful and failed communication.

While our core contribution is a technical investigation of AI-to-AI communication under perceptual mismatch, the fundamental motivation for modeling divergent visual perception stems from the critical need for accessible and inclusive AI systems. Visual impairments are a widespread global health issue, with an estimated 2.2 billion people living with vision impairment or blindness according to the World Health Organization (WHO) World Health Organization & The Lancet Global Health Commission on Global Eye Health (2021). The sheer scale of this population underscores the necessity for LLMs to communicate effectively with users whose visual experiences differ significantly from the "normal" vision assumed by most models. Ensuring that LLMs can adapt their communication to users with divergent perceptual abilities is not merely a matter of engineering efficiency, but a prerequisite for deploying equitable, human-centric AI that serves the entire global population.

In summary, our contributions are:

- We introduce a new setting for studying agent alignment under divergent visual perceptions, simulating distortions inspired by real human visual impairments.
- We conduct a systematic analysis of agent adaptation across both offline (SFT, DPO) and online (KTO, GRPO) learning paradigms.
- We release a preference dataset, constructed from distorted and undistorted inputs, to support future research on multimodal agent collaboration.

## 2 RELATED WORK

Human communication often requires adapting to partners with different perceptual experiences, for example in cases of impaired or limited vision. Inspired by this, recent work has introduced multimodal reference games as testbeds for studying alignment (Corona et al., 2019; Takmaz et al., 2023) between communicating agents. However, most approaches still assume identical perceptual inputs across agents, overlooking the mismatches that frequently arise in practice due to sensory limitations or hardware degradation.

Personalization in language models has long been studied, particularly in dialogue systems (Serban et al., 2015; Song et al., 2019; Zhang et al., 2019). Some Theory of mind (ToM)-based approaches (Ma et al., 2023) propose plug-and-play modules that update weights dynamically (Takmaz et al., 2023) or model listener behavior internally (Raileanu et al., 2018). While prior work has explored speaker–listener adaptation in text-only settings (Wang et al., 2024a), we extend this line to multimodal image reference games. Other personalization methods learn on large scale user dialogue histories (Ma et al., 2021; Zhong et al., 2022), whereas our focus is on both online and offline adaptation strategies.

Parameter-efficient fine-tuning enables scalable adaptation of multimodal models. LoRA modules (Hu et al., 2022a) add trainable low-rank adapters on top of frozen backbones and have inspired many extensions (Zhang et al., 2023; Lialin et al., 2023; Liu et al., 2023; Wu et al., 2024; Sheng et al., 2023; yang Liu et al., 2024). Alternative methods adapt only small subsets of weights (Ben Zaken et al., 2022; Ansell et al., 2021) or use adapters embedded within or alongside network layers (Pfeiffer et al., 2020; Sung et al., 2022; Mercea et al., 2024). In this work, we adopt widely used LoRA for adapting MLLMs.

Reinforcement learning provides another route for adaptation, especially in online settings (Snell et al., 2023; Ziegler et al., 2019; Ramamurthy et al., 2023). GRPO (Shao et al., 2024) stabilizes training by normalizing rewards within groups, avoiding the need for a critic and reducing variance. Preference-based methods such as KTO (Ethayarajh et al., 2024) and DPO (Rafailov et al., 2024) optimize directly from positive/negative feedback pairs. Related efforts (Guo et al., 2024; Liu et al., 2024a) explore online adaptation via model feedback, but our work focuses specifically on aligning communication under mismatched perceptual conditions.

Visual reference identification has been studied in multimodal dialogue (de Vries et al., 2016; Ni et al., 2021; Alaniz et al., 2021; Das et al., 2016). While Corona et al. (2019) examined attribute-based reference games, we broaden the scope to free-form description generation in the presence of perceptual impairments. One of our contribution is to systematically compare online and offline adaptation strategies in this setting.

## 3 Aligning Agents with Divergent Visual Spaces

### 3.1 Task Formulation

We study a variant of the image reference game where speaker and listener have divergent visual spaces. Both agents are given a target-confounding image pair, but the *speaker* observes the undistorted pair $(x_{\text{tgt}}, x_{\text{conf}})$, presented sequentially as a left and right image, and produces a message $m$ describing the target. The *listener*, in contrast, receives only the distorted pair $(\hat{x}_{\text{tgt}}, \hat{x}_{\text{conf}})$ in random order and must identify the target image from this pair conditioned on $m$. The listener chooses an action $a \in \{\text{left}, \text{right}, \text{none of these}\}$, corresponding to selecting the left image, the right image, or rejecting both as inconsistent with the message. Success is achieved when $a$ points to the target (e.g., $a = \text{left}$ when the left image is $\hat{x}_{\text{tgt}}$).

The $i$-th interaction yields a binary reward $r_i \in \{0, 1\}$, where $r_i = 1$ if the listener correctly identifies the target and $r_i = 0$ otherwise. If the listener selects "none of these," the game is treated as a failure ($r_i = 0$). Because the speaker has access to richer visual information than the listener, effective communication requires generating descriptions that remain faithful under distortion, avoiding reliance on perceptual details invisible to the listener. This framework is further depicted in Fig. 1.

### 3.2 Perceptual Distortions

We design five perceptual distortions inspired by real human visual impairments, applied in either the spatial or pixel domain. First, cataract, a condition where the eye's lens becomes clouded and vision turns blurry with reduced contrast, is simulated via Gaussian blurring coupled with contrast reduction, after which random black artifacts are added to the image. Second, age-related macular degeneration (AMD), which leads to the deterioration of the central retina and hampers fine-detail recognition while leaving peripheral vision intact, is modeled by masking the central region of the image. Third, color blindness, a deficiency in perceiving chromatic information, is represented by removing color channels and converting to luminance-only. Fourth, tunnel vision, typically caused by glaucoma or retinitis pigmentosa and characterized by the loss of peripheral vision, is simulated by masking out peripheral regions to retain only a central circular window. Lastly, retinal detachment, where the retina separates and creates dark, irregular blind spots often starting at the periphery or lower visual field, is simulated by overlaying irregular dark patches on the lower half of the image. An example of these perceptual distortions can be seen in Fig. 1. For further details, we refer the reader to Appendix E.

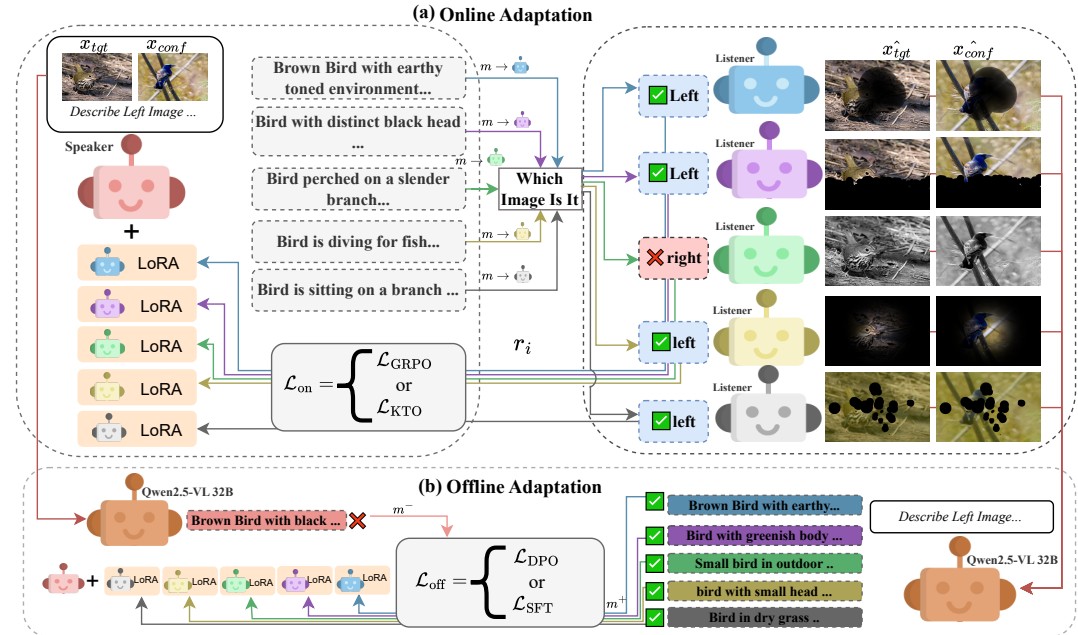

Figure 1: **Overview of the framework.** (a) The speaker, adapted with LoRA, generates descriptions for the target image and updates online via reinforcement learning (GRPO or KTO) based on listener feedback. Listeners receive the description with distorted images and predict the referred image, providing rewards to the speaker. (b) An offline data generation pipeline creates labeled descriptions for offline adaptation with supervised finetuning and direct preference optimization (Rafailov et al., 2023). LoRA modules are trained for each distortion to adapt the speaker.

### 3.3 PREFERENCE DATASET CONSTRUCTION

To enable offline adaptation in the reference game with divergent visual spaces, we construct a preference dataset using Qwen2.5-VL-32B. The central idea is to create contrasting descriptions from conditioning on distorted versus undistorted image pairs (Fig. 1). Distorted-view descriptions serve as positive preferences, since they are aligned with the listener's perceptual limitations, while undistorted-view descriptions serve as negative preferences, as they often rely on features inaccessible under distortion. This preference framing explicitly encodes communicative success under mismatched perceptual inputs.

We generate paired descriptions $(m^+, m^-)$ with Qwen2.5-VL-32B where $m^+$ represents positive preference and $m^-$ negative preference. The model is always presented with two images in order *Image 1* (target) and *Image 2* (confounding) and is instructed to describe only *Image 1*. For the distorted-view query, we provide $(\hat{x}_{tgt}, \hat{x}_{conf})$ and record the output as $m^+$; for the normal-view query, we provide $(x_{tgt}, x_{conf})$ and record the output as $m^-$. Keeping the instruction prompt constant ensures that differences in responses arise solely from the change in visual input. By construction, $m^+$ captures distortion-consistent descriptions interpretable under impaired perception, while $m^-$ encodes distortion-inconsistent content. This yields preference pairs labeled $m^+ \succ m^-$, providing the contrastive supervision required by preference-based alignment methods such as DPO. A final training instance from this preference dataset then contains the undistorted images $(x_{tgt}, x_{conf})$, the positive preference $m^+$, and the negative preference $m^-$.

## 3.4 DATASET CONSTRUCTION

For each target image $x_{\text{tgt}}$, we randomly sample a confounding image $x_{\text{conf}}$ from the same dataset to act as a distractor, without applying any post-processing or semantic filtering. Each pair is instantiated under both normal and distorted views, yielding the tuple

$$((x_{\text{tgt}}, x_{\text{conf}}), (\hat{x}_{\text{tgt}}, \hat{x}_{\text{conf}})), \texttt{ instruction}, (m^+, m^-)),$$

where $(\hat{x}_{\text{tgt}}, \hat{x}_{\text{conf}})$ denote their distorted counterparts. Qwen2.5-VL-32B is queried twice per instance—once with distorted inputs to produce $m^+$ and once with clean inputs for $m^-$—using identical text prompts and a fixed image ordering that always places the target as *Image 1*. This ensures that all variation in descriptions arises solely from differences in visual perception. The unified prompt template is:

> You see two images. Image 1: [IMG_1], Image 2: [IMG_2]. Focus on Image 1. Your goal
> is to describe Image 1 in a way that clearly distinguishes it from Image 2. Do not mention
> Image 2 in your description at all. Highlight unique features of Image 1 that differentiate it
> from Image 2. Ignore the

Under distortion, this template promotes robustness by encouraging features that remain interpretable despite perceptual noise, while strictly prohibiting reference to the distractor. The model thus produces paired descriptions $(m^+, m^-)$ that encode a binary preference $m^+ \succ m^-$ consistent with distortion-aware communication, supporting preference-based optimization such as DPO. All instances are tokenized with a 2048-token context limit to preserve the full multimodal structure. The resulting dataset provides high-quality contrastive supervision, aligning speaker behavior with the perceptual constraints of an impaired listener.

## 3.5 ADAPTING ALGORITHMS

We study offline and online post-training methods that are complementary in cost, stability, and feedback needs. Offline adaptation (SFT, DPO) is a cost-effective ante-hoc strategy when perceptual distortion is known in advance; SFT is a strong baseline and DPO (Rafailov et al., 2023) is a widely adopted preference-optimization method that avoids training a reward model while enforcing a KL constraint. Online adaptation (KTO (Ethayarajh et al., 2024), GRPO (Shao et al., 2024)) captures in situ personalization when adaptation to individual listeners is desired and the visual divergence is unknown. All methods are applied to the speaker policy $\pi_\theta$ via LoRA (Hu et al., 2022b).

For offline adaptation, we train on preference triples $\{(\mathbf{x}, m^+, m^-)\}$ where $\mathbf{x} = (x_{\text{tgt}}, x_{\text{conf}})$ is the undistorted input pair, $m^+$ is the positive preference, and $m^-$ is the negative preference. All algorithms condition on $\mathbf{x}$. SFT fine-tunes the speaker directly on positive responses $m^+$, providing a simple baseline for offline adaptation. DPO leverages paired preferences $(m^+, m^-)$ to optimize directly without learning a reward model, offering stability and efficiency for offline adaptation. For online adaptation, we employ KTO and GRPO. KTO incorporates principles from behavioral economics, capturing asymmetries such as loss aversion, which makes it well-suited for online adaptation with noisy or uneven feedback. GRPO is a critic-free reinforcement learning method that normalizes rewards across sampled groups, enabling stable and scalable online alignment in interactive settings. In summary, we study two offline objectives $\mathcal{L}_{\text{off}} = \{\mathcal{L}_{\text{SFT}}, \mathcal{L}_{\text{DPO}}\}$ and two online objectives $\mathcal{L}_{\text{on}} = \{\mathcal{L}_{\text{KTO}}, \mathcal{L}_{\text{GRPO}}\}$, as depicted in Fig. 1. Details of algorithms are in Appendix G.

## 4 RESULTS

### 4.1 DATASETS AND EVALUATION PROTOCOL

We evaluate adaptation on two benchmarks: **CLEVR** (Johnson et al., 2017), which contains synthetic scenes with objects varying in shape, color, material, and size and stresses compositional reasoning, and **CUB** (Wah

et al., 2011), which consists of real bird species images and tests fine-grained visual recognition. For each dataset, we generate paired samples for the reference game, presenting the speaker with undistorted images and the listener with its distorted counterparts.

For checkpoint selection, we adopt two evaluation protocols. First, we select the checkpoint with the highest test accuracy (*Test-Pick*), representing an optimistic upper bound due to the high variance of RL training. Second, we select the checkpoint with the highest validation accuracy under greedy sampling (*Val-Pick*), which provides a more realistic model selection strategy. For both checkpoints, we evaluate using nucleus sampling with temperature 0.7 and report results averaged over 10 independent runs on the same test datset. All models are adapted by tuning LoRA adapters on the speaker side, while the listener model remains fixed. For extended quantitative results, we refer the reader to the Appendix A.

### 4.2 QUANTIATIVE RESULTS

We report results on CLEVR and CUB in Tab. 1 and Tab. 2. Each distortion poses distinct challenges in the zero-shot setting (ZS) prior to any adaptation: AMD and cataract obscure central or global structure, grayscale removes color cues critical for fine-grained recognition, tunnel vision limits peripheral context, and detached retina introduces blurring.

On CLEVR, all adaptation methods improve over the ZS baseline in average test-pick accuracy, but trade-offs between test-pick and val-pick performance remain clear. KTO shows the largest overall gains, with a remarkable +33.6% improvement on grayscale (maximum) and a stable +6.8% average improvement under validation checkpoints. GRPO provides smaller but consistent improvements (+5.1% on grayscale and +5.1% on detached retina) with modest averages (+2.5% maximum, +2.7% validation), making it conservative but reliable. DPO achieves the highest average maximum gain (+8.3%) with strong improvements on grayscale (+27.5%) but collapses under validation-based evaluation (+0.04%). SFT demonstrates strong peaks on AMD (+14.8%) but suffers instability by loosing 2.4% in val-pick accuracy. SFT shows decent improvements in most distortions except Grayscale where it looses 4% and reports a modest average val-pick gain of 0.51%. We believe AMD presents a particularly challenging distortion, as the object of interest is often majorly or even fully occluded, which increases the model's uncertainty and limits its ability to generate reliable descriptions meaningful background information.

On CUB, all methods surpass the ZS baseline, though improvements are smaller than on CLEVR due to the already high starting accuracy. KTO and GRPO are the most consistent, with KTO yielding strong gains on cataract (+7.2 Test-Pick, +2.2 Val-Pick) and AMD (+17.7), leading to the best overall averages (+5.8 / +4.2). GRPO is similarly competitive, with the strongest AMD improvement (+18.2) and stable detached retina gains (+4.1), resulting in +5.5 / +3.7 average boosts. DPO achieves the highest Test-Pick on grayscale (+11.1) but collapses on detached retina (−10.0), leaving it with only +0.9 / −1.0 average gains. SFT performs reliably across most distortions, especially grayscale (+5.7 / +7.6) and AMD (+9.3), with balanced improvements overall (+4.0 / +3.1).

In summary, KTO remains the most reliable online method across both benchmarks, providing the strongest and most stable improvements. GRPO emerges as a consistent competitor and is outperformed by DPO in average test pick accuracy on CLEVR. Offline methods (DPO, SFT) continue to achieve the significant peak improvements but exhibit poor generalization when checkpoint selection is validation-based, highlighting instability in practical use.

### 4.3 QUALITATIVE RESULTS

We present qualitative examples in Fig. 2 to highlight key observations. In each case, the left column shows the original images $x$, while the rightmost column shows its distorted counterparts $\hat{x}$. For both, we report the base model's description and the adapted model's description. Method names are shown in blue, text in red indicates changes relative to the base description, and **green text** marks newly introduced information.

| Method | Cataract | Grayscale | Tunnel Vision | Detached Retina | AMD | Avg. |
|--------|----------|-----------|---------------|-----------------|-----|------|
| ZS | 60.5 | 39.2 | 68.9 | 78.9 | 75.7 | 64.64 |
| KTO | 63.4/62.3 (+2.9/+1.8) | **72.8/75.6** (+33.6/+36.4) | **75.6**/70.1 (+6.7/+1.2) | 79.8/81.1 (+0.9/+2.2) | 69.7/68.1 (-6.0/-7.6) | 72.26/71.44 (+7.62/+6.80) |
| GRPO | 61.3/61.3 (+0.8/+0.8) | 44.3/44.2 (+5.1/+5.0) | 75.1/73.2 (+6.2/+4.3) | 84.0/84.0 (+5.1/+5.1) | 70.8/74.2 (-4.9/-1.5) | 67.10/67.38 (+2.46/+2.74) |
| DPO | 62.6/60.6 (+2.1/+0.1) | 66.7/39.9 (+27.5/+0.7) | 75.4/68.8 (+6.5/-0.1) | **84.5**/79.6 (+5.6/+0.7) | 75.6/74.1 (-0.1/-1.6) | **72.96**/64.60 (+8.32/-0.04) |
| SFT | **66.6**/61.73 (+6.1/+1.23) | 35.2/39.5 (-4.0/+0.3) | 71.0/71.0 (+2.1/+2.1) | 80.7/80.2 (+1.8/+1.3) | **90.5**/73.3 (+14.8/-2.4) | 68.80/65.15 (+4.16/+0.51) |

Table 1: Adaptation performance under different visual distortions for **CLEVR** (Johnson et al., 2017). We use Qwen2.5-VL 7B as speaker and listener. For each method we report two test accuracies: Test-Pick Accuracy / Val-Pick Accuracy, with gains relative to **ZS**. Positive gains are in green, negatives in red. Best numbers per column (Max) are in **bold**. The last column reports average accuracies across distortions for both metrics, with gains relative to the ZS average.

| Method | Cataract | Grayscale | Tunnel Vision | Detached Retina | AMD | Avg. |
|--------|----------|-----------|---------------|-----------------|-----|------|
| ZS | 73.7 | 75.4 | 91.2 | 90.0 | 76.6 | 81.38 |
| KTO | **80.9**/75.9 (+7.2/+2.2) | 72.9/72.8 (-2.5/-2.6) | 94.6/91.6 (+3.4/+0.4) | 93.2/93.2 (+3.2/+3.2) | 94.3/94.3 (+17.7/+17.7) | **87.18**/85.56 (+5.80/+4.18) |
| GRPO | 75.7/73.1 (+2.0/-0.6) | 75.0/74.7 (-0.4/-0.7) | **94.9**/90.9 (+3.7/-0.3) | **94.1/94.1** (+4.1/+4.1) | **94.8**/92.4 (+18.2/+15.8) | 86.90/85.04 (+5.52/+3.66) |
| DPO | 76.2/74.4 (+2.5/+0.7) | **86.5**/78.8 (+11.1/+3.4) | 93.7/93.7 (+2.5/+2.5) | 80.0/80.0 (-10.0/-10.0) | 75.1/75.0 (-1.5/-1.6) | 82.30/80.38 (+0.92/-1.00) |
| SFT | 74.0/74.1 (+0.3/+0.4) | 81.1/83.0 (+5.7/+7.6) | 93.6/94.6 (+2.4/+3.4) | 92.5/92.5 (+2.5/+2.5) | 85.9/78.0 (+9.3/+1.4) | 85.42/84.4 (+4.04/+3.06) |

Table 2: Adaptation performance under different visual distortions for **CUB** (Wah et al., 2011). We use Qwen2.5-VL 7B as speaker and listener. For each method we report two test accuracies: Test-Pick Accuracy / Val-Pick Accuracy, with gains relative to **ZS**. Positive gains are in green, negatives in red. Best numbers per column (Max) are in **bold**. The last column reports average accuracies across distortions for both metrics, with gains relative to the ZS average.

Adapted models generally produce descriptions that are more consistent with the distorted view, leading to improved speaker–listener communication. For AMD distortion, adapted model enriches the description by emphasizing fine details such as the bird's beak and the rocky surface. Under Detached Retina, adapted model incorporates contextual background cues, describing the blurred foliage, which provides the listener with disambiguating information. In the Grayscale case, the adapted model avoids reliance on color-based cues and instead introduces structure-sensitive descriptors such as "a thin, bare branch," which align more closely with the listener's visual perception. For Tunnel Vision, the adapted model improves the base description "just caught a fish" to the more faithful "dives towards the water," aligning with what the listener actually sees, where the fish is not clearly visible. Finally, in the Cataract setting, adapted model strengthens the description of the target bird while adding background details about vegetation and posture, helping the listener distinguish the target from confounding images. Overall, these examples demonstrate how adaptation enables the speaker to better align with the listener's distorted visual space, effectively allowing both to "see the same picture." These qualitative findings are consistent with our quantitative results, showing that adaptation improves interpretability and fidelity as well.

## 5 ROBUSTNESS ANALYSIS

### 5.1 CROSS DATASET TRANSFER

We next evaluate the cross-dataset generalization of adaptation methods by training on one dataset and testing on the other. Specifically, we first train on CLEVR and evaluate on CUB (Table 3). We then mirror this procedure by training on CUB and evaluating on CLEVR (Table 4). for evaluation, we report test-pick accuracy as mentioned in section. 4.1. In each case, we report results for KTO, GRPO, DPO, and SFT under three representative distortions (cataract, grayscale, tunnel vision).

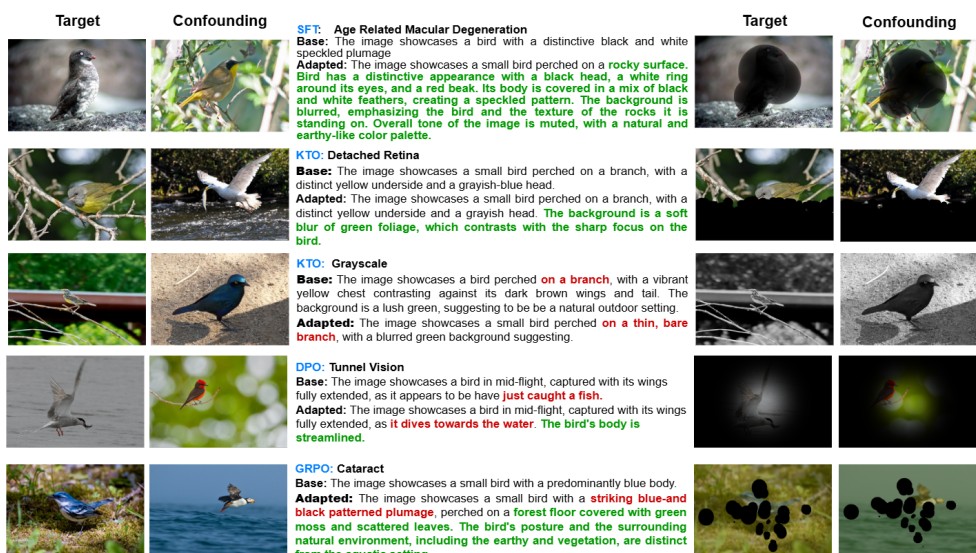

Figure 2: **Qualitative results** for Qwen2.5-VL 7B as speaker. The left image is being described in each description. On left we share original images and on right we share distorted images. In each description text with color blue represents the method name. **Red text** represent change of information from original description while **green text** represents new information.

In the CLEVR→CUB setting, adaptation consistently improves over ZS across all methods in average accuracy. KTO yields balanced improvements, particularly on cataract (+4.4) and tunnel vision (+3.0), averaging +3.0 overall. GRPO shows similar behavior, with moderate gains (+2.0 cataract, +1.9 grayscale) for an average of +1.4. DPO underperforms on cataract (–1.5) but achieves the highest boost on grayscale (+5.8). SFT demonstrates the strongest distortion-specific generalization, with large improvements on cataract (+5.2) and especially grayscale (+14.1), resulting in the highest average (+6.5). These results highlight that while online methods transfer more stably, SFT can achieve the strongest cross-dataset gains in this direction.

In contrast, the CUB→CLEVR transfer proves more challenging, with smaller and more variable improvements. KTO achieves modest average gains (+2.4), driven by tunnel vision (+6.9) despite a drop on grayscale (–1.5). GRPO follows a similar pattern, excelling on tunnel vision (+10.7) but degrading on grayscale (–2.7), yielding +3.4 on average. SFT generalizes most reliably, with improvements on grayscale (+4.4) and tunnel vision (+1.1), though average gains remain modest (+1.8). DPO lags behind in cataract while reporting a considerable gain of on tunnel vision (+4.4) and achieving an average gain of +4.1. Overall, we observe that compositional nature of CLEVR dataset makes transfer from CLEVR to natural-image CUB is more successful, while the reverse direction remains more challenging, with KTO showing more stable improvements among online methods.

## 5.2 ADAPTING TO DIFFERENT AGENT UNDER DIVERGENT VISUAL SPACE

Table 5 reports adaptation results on the CUB dataset using Qwen2.5-VL-7B as the speaker and InternVL3.5 (Wang et al., 2025) as the listener. The baseline listener is already strong, with near-ceiling accuracy on Tunnel Vision (96.0) and Detached Retina (97.3), but weaker under Cataract (80.4), Grayscale (72.0), and AMD (81.7). In this high-performing regime, adaptation provides only modest gains. KTO improves slightly on Grayscale (+1.9 Test-Pick), matches or marginally improves on Tunnel Vision and AMD,

| Method | Cataract | Grayscale | Tunnel Vision | Avg. |
|--------|----------|-----------|---------------|------|
| ZS | 73.7 | 75.4 | 91.2 | 80.1 |
| KTO | 78.1 (+4.4) | 77.1 (+1.7) | **94.2** (+3.0) | 83.1 (+3.0) |
| GRPO | 75.7 (+2.0) | 77.3 (+1.9) | 91.6 (+0.4) | 81.5 (+1.4) |
| DPO | 72.2 (-1.5) | 81.2 (+5.8) | 91.9 (+0.7) | 81.8 (+1.7) |
| SFT | **78.9** (+5.2) | **89.5** (+14.1) | 91.5 (+0.3) | **86.6** (+6.5) |

Table 3: Adaptation performance for **CLEVR→CUB**. We use Qwen2.5-VL 7B as speaker and listener. We report ZS accuracy and Test-Pick accuracy (gains in green/red). Best results per distortion are in **bold**. The last column shows the row-wise average across distortions.

| Method | Cataract | Grayscale | Tunnel Vision | Avg. |
|--------|----------|-----------|---------------|------|
| ZS | 60.5 | 39.2 | 68.9 | 56.2 |
| KTO | 62.4 (+1.9) | 37.7 (-1.5) | 75.8 (+6.9) | 58.6 (+2.4) |
| GRPO | **62.6** (+2.1) | 36.5 (-2.7) | **79.6** (+10.7) | 59.6 (+3.4) |
| DPO | 57.8 (-2.7) | **50.5** (+11.3) | 72.7 (+3.8) | **60.3** (+4.1) |
| SFT | 60.4 (-0.1) | 43.6 (+4.4) | 70.0 (+1.1) | 58.0 (+1.8) |

Table 4: Adaptation performance for **CUB→CLEVR**. We use Qwen2.5-VL 7B as speaker and listener. We report ZS accuracy and Test-Pick accuracy (gains in green/red). Best results per distortion are in **bold**. The last column shows the row-wise average across distortions.

| Method | Cataract | Grayscale | Tunnel Vision | Detached Retina | AMD | Avg. |
|--------|----------|-----------|---------------|-----------------|-----|------|
| ZS | 80.4 | 72.0 | 96.0 | 97.3 | 81.7 | 85.5 |
| KTO | **79.0**/78.9 (-1.4/-1.5) | **73.9**/72.1 (+1.9/+0.1) | 96.0/**96.2** (+0.0/+0.2) | 95.4/**96.9** (-1.9/-0.4) | 93.3/93.0 (+11.6/+11.3) | **87.5**/87.4 (+2.0/+1.9) |
| GRPO | 76.4/77.8 (-4.0/-2.6) | 72.5/70.7 (+0.5/-1.3) | 95.1/95.3 (-0.9/-0.7) | 96.0/94.9 (-1.3/-2.4) | 93.0/**93.5** (+11.3/+11.8) | 86.6/86.4 (+1.1/+0.9) |

Table 5: Adaptation performance under different visual distortions with InternVL3.5 (Wang et al., 2025) as listener for **CUB** (Wah et al., 2011). We use Qwen2.5-VL 7B as speaker. We report *Test-Pick/Val-Pick* accuracies with gains relative to **ZS**. Positive gains are in green, negatives in red. Best adapted values per column are in **bold**. The last column reports row-wise averages (and gains) across distortions.

but underperforms on Cataract and Detached Retina, averaging 87.52/87.42 (+2.0/+1.9). GRPO achieves the strongest boost on AMD (+11.3/+11.8) but falls behind on Cataract and Tunnel Vision, yielding slightly lower averages (86.6/86.4). Overall, gains are limited, reflecting that adaptation has less room to help when the baseline listener already performs well. Furthermore, the weaker results on Cataract and Detached Retina highlight the challenges of adapting across heterogeneous agents.

Table 6 presents results on CLEVR with the same speaker–listener pairing. Here the baseline is much weaker overall (51.6 average), and especially poor on Grayscale (27.1). In this lower-performing regime, adaptation produces substantial improvements. KTO provides the most consistent and significant gains, especially on Grayscale (+34.4/+33.2), with improvements also on Cataract and Tunnel Vision, raising the averages to 58.9/58.9 (+7.3/+7.3). GRPO offers a smaller benefit, with strong improvements under Tunnel Vision (+5.7/+2.3) and Detached Retina (+3.1/–0.2), but lags on Cataract and Grayscale, resulting in averages of 52.7/51.2 (+1.1/–0.5).

Taken together, these findings reveal a sharp asymmetry. When the baseline listener is already strong, as in CUB, adaptation provides only minor and uneven gains. By contrast, when the listener is weaker, as in CLEVR, adaptation—especially with KTO—yields large and reliable improvements. Compared to earlier results with Qwen2.5-VL-7B as both speaker and listener, these results suggest that adapting across heterogeneous agents is more challenging, with smaller benefits for strong listeners but clear advantages when the listener struggles under perceptual divergence.

# 6 ABLATION: HETEROGENEOUS MODEL SIZES

In this ablation, we investigate how heterogeneous speaker–listener model sizes affect performance on the CUB dataset using KTO adaptation. We evaluate four configurations combining Qwen-2.5 VL 3B and 7B models in both symmetric (3B/3B, 7B/7B) and asymmetric (3B/7B, 7B/3B) roles. Zero-shot (Base) and post-adaptation Max accuracies are reported in Tab. 7. A key observation is that the largest symmetric setup

| Method | Cataract | Grayscale | Tunnel Vision | Detached Retina | AMD | Avg. |
|--------|----------|-----------|---------------|-----------------|-----|------|
| ZS | 52.0 | 27.1 | 55.4 | 60.9 | **62.7** | 51.6 |
| KTO | **52.1**/51.4 (+0.1/-0.6) | **61.5**/60.3 (+34.4/+33.2) | 57.8/59.5 (+2.4/+4.1) | 61.4/61.4 (+0.5/+0.5) | 61.9/61.9 (-0.8/-0.8) | **58.9**/**58.9** (+7.3/+7.3) |
| GRPO | 51.1/51.1 (-0.9/-0.9) | 26.5/25.6 (-0.6/-1.5) | **61.1**/57.7 (+5.7/+2.3) | **64.0**/60.7 (+3.1/-0.2) | 60.7/60.7 (-2.0/-2.0) | 52.7/51.2 (+1.1/-0.5) |

Table 6: Adaptation performance under different visual distortions with InternVL3.5 (Wang et al., 2025) as listener for CLEVR (Johnson et al., 2017) dataset. We use Qwen2.5-VL 7B as speaker. We report Test-Pick/Val-Pick accuracies (with gains in green for improvements and red for drops). Best adapted values per column are in **bold**. The last column reports row-wise averages with gains relative to the ZS average.

| Method | Model Configuration | | Type | Accuracy [%] | | | | |
|--------|---------|----------|------|-----|----------|-----------------|-----------|---------|
| | Speaker | Listener | | AMD | Cataract | Detached Retina | Grayscale | Average |
| **KTO** | Qwen-2.5 VL 3B | Qwen-2.5 VL 3B | Base | 88.0 | 67.0 | 96.0 | 94.0 | 86.3 |
| | | | Max | 90.0 (+2.0) | 70.0 (+3.0) | 100.0 (+4.0) | 98.0 (+4.0) | 89.5 (+3.2) |
| | Qwen-2.5 VL 3B | Qwen-2.5 VL 7B | Base | 94.0 | 80.0 | 98.0 | 80.0 | 88.0 |
| | | | Max | 96.0 (+2.0) | 85.0 (+5.0) | 99.0 (+1.0) | 82.0 (+2.0) | 90.5 (+2.5) |
| | Qwen-2.5 VL 7B | Qwen-2.5 VL 3B | Base | 92.0 | 67.0 | 96.0 | 93.0 | 87.0 |
| | | | Max | 93.0 (+1.0) | 75.0 (+8.0) | 97.0 (+1.0) | 97.0 (+4.0) | 90.8 (+3.8) |
| | Qwen-2.5 VL 7B | Qwen-2.5 VL 7B | Base | 90.0 | 63.0 | 76.0 | 88.0 | 81.4 |
| | | | Max | 94.0 (+4.0) | 81.0 (+18.0) | 79.0 (+3.0) | 93.0 (+5.0) | 87.8 (+6.4) |

Table 7: Zero-shot (Base) and maximum KTO accuracy across distortions on CUB.

(7B/7B) does *not* produce the strongest baseline: it has the lowest average Base accuracy (81.4%), driven by substantial drops under Cataract and Detached Retina distortions. In contrast, the asymmetric 3B speaker / 7B listener setup yields the best baseline (88.0%), indicating that a stronger listener is crucial for reliably interpreting the speaker's descriptions, even when the speaker is small. Adaptation with KTO consistently improves performance across all configurations. The largest gains occur in the 7B/7B setup, which benefits the most from correction of its weak baseline, particularly on Cataract (+18 points). However, the configuration achieving the highest *final* accuracy is again the asymmetric 3B/7B pair, reaching 90.5% after KTO. This highlights an efficient design choice: pairing a lightweight speaker with a stronger listener offers a favorable balance between computational cost and top-end performance. We share extended analysis of in section B

## 7 CONCLUSION

We studied aligning multimodal agents that inhabit *divergent visual spaces*. Using an image reference game with five distortion types, we compared offline (SFT, DPO) and online (KTO, GRPO) post-training strategies. Our experiments show that (i) offline adaptation can help without access to the listener when provided with high-quality preference pairs aligned to the listener's impairment; (ii) online adaptation with KTO and GRPO delivers strong improvements across distortions by learning directly from interaction feedback; and (iii) qualitative analyses confirm that adapted speakers prioritize distortion-consistent cues, bridging mismatches in perception. Cross-dataset studies further indicate that compositional nature of CLEVR dataset makes it easier to transfer to natural images (CUB). Adapting to a different listener family yields smaller gains, underscoring the challenge of alignment across heterogeneous model representations.

## REPRODUCIBILITY STATEMENT

We will release all artifacts needed to replicate our results: the full training/evaluation code for offline (SFT, DPO) and online (KTO, GRPO) adaptation with LoRA; scripts to generate and apply our five visual distortions; data preparation/evaluation pipelines for the image reference game; and a public preference dataset containing instructions and paired $(m^+, m^-)$ descriptions. We document exact base/model versions (e.g., Qwen2.5-VL-7B/32B, InternVL3.5) wherever they are used and in Appendix F. We share dataset generation details in section 3.3. Our main reported numbers use nucleus sampling with temperature 0.7 averaged over multiple runs; greedy-decoding results (max test accuracy and validation-picked checkpoints) are included in the appendix for transparency in section A.

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
