**Appendix Contents**

- Extended Quantitative Results
    - Tables: CLEVR results, CUB results, ImageNet results
    - Ablation: Heterogeneous Model Sizes
        * Table: Speaker–listener size configurations
- Perceptual Distortions
    - Figure: Examples of visual distortions
- Prompt Template Ablations
    - Ablation over prompt template
        * Table: Distortion-aware prompting vs. zero-shot
        * Table: Qualitative cataract-aware generations (CLEVR)
- Performance of Top-Tier Models
    - Table: Top-tier speakers on ImageNet
    - Table: Top-tier speakers on CLEVR
- Hyperparameters
    - Dataset-level defaults (CUB, CLEVR, ImageNet)
    - Method-specific settings (GRPO, DPO, KTO, SFT)
- Adapting Algorithms
    - Supervised Fine-Tuning (SFT)
    - Direct Preference Optimization (DPO)
    - Kahneman–Tversky Optimization (KTO)
    - Group Relative Policy Optimization (GRPO)
- Extended Qualitative Results
    - Figure: GRPO qualitative examples
    - Figure: KTO qualitative examples
    - Figure: DPO qualitative examples
    - Failure Cases
        * Figure: Failure cases where adapted model fails but base succeeds
- Datasets
    - CLEVR (Johnson et al., 2017)
    - CUB (Wah et al., 2011)
- Linguistic Analysis
    - Shift in term frequencies under grayscale distortion
        * Figure: Category-wise term frequencies (base vs. adapted)
    - Quantitative linguistic shifts across distortions
        * Figure: KTO – word count and Flesch Reading Ease
        * Figure: GRPO – word count and Flesch Reading Ease
- Theoretical Analysis

SUPPLEMENTARY MATERIAL

This supplementary document provides additional details and results complementing the main paper. We begin by reporting extended quantitative results on both **CLEVR** (Johnson et al., 2017) and **CUB** (Wah et al., 2011), and report maximum test accuracy and validation-set–picked checkpoints under greedy sampling. We further present qualitative examples that illustrate how adaptation strategies manifest across diverse perceptual distortions, offering insight into the interpretability and alignment of generated descriptions.

To ensure transparency and reproducibility, we also describe our experimental setup in more depth, covering dataset construction, preference pair generation, hyperparameters, and implementation details for each adaptation algorithm. Finally, we provide extended visualizations of the distortions used in our experiments, as well as additional qualitative examples for both online and offline adaptation methods.

## A  EXTENDED QUANTIATIVE RESULTS

We report results on CLEVR (Johnson et al., 2017) and CUB (Wah et al., 2011) in Tab. 9 and Tab. 10 for maximum test accuracy and test accuracy of validation-set-picked checkpoint under greedy sampling.

On CLEVR, all adaptation methods improve over the ZS baseline. KTO (Ethayarajh et al., 2024) shows the strongest overall improvements, with a remarkable +42% gain under grayscale (maximum) and a stable +6.4% average improvement under validation-set-picked checkpoints. GRPO (Shao et al., 2024) produces smaller peak gains (+13% on grayscale) but demonstrates more stable validation-set-picked checkpoint's performance (+2.8% average), making it conservative but reliable. Among offline methods, DPO (Rafailov et al., 2023) achieves the highest average maximum gain (+10.8%) but collapses to only +0.8% under validation-set-picked checkpoint's evaluation, revealing instability. SFT attains strong peaks on AMD (+15%) and tunnel vision (+11%), but its validation-set-picked checkpoint's scores collapse on cataract (–11%), dragging its average below baseline (–1.2%).

On CUB, improvements are smaller. KTO again proves robust, yielding strong gains on cataract (+18.0 maximum, +13.0 for validation-set-picked checkpoint) and detached retina (+5.0), leading to average improvements of +6.4% (maximum) and +5.0% (validation-set-picked checkpoint). GRPO is highly competitive, achieving the highest overall averages (+7.0% maximum, +4.6% validation-set-picked checkpoint), with especially strong performance on cataract (+19.0 / +15.0). Among offline methods, DPO achieves the largest maximum gains (+8.8% average), driven by cataract (+26.0) and grayscale (+9.0), while SFT excels under tunnel vision and detached retina (+6.0 to +8.0). However, both offline methods drop significantly when evaluated with validation-set-picked checkpoint checkpoints (–7.0 to –8.0), again highlighting instability under practical selection.

Finally, on ImageNet (Tab. 8), the high ZS baseline of 83.0% leads to more modest but consistent gains across methods. DPO achieves the highest average maximum accuracy (88.0%) and gain (+5.0%), with strong performance on Grayscale (+5.0%) and Tunnel Vision (+7.0%). GRPO stands out for its reliability, showing the most stable performance with almost no drop between maximum and validation-picked scores (87.6% vs. 87.4%) and delivering a solid average gain of +4.4%. SFT shows strong peak performance on specific distortions like Cataract (+9.0%) and achieves the best scores on AMD and Detached Retina, but its zero gain on Grayscale and Tunnel Vision limits its average improvement. KTO provides a modest but stable average gain (+1.0% for validation-picked) but is uniquely penalized by the Grayscale distortion, where it underperforms the ZS baseline by -4.0%. Overall, on this challenging benchmark, all methods demonstrate positive average gains for validation-picked checkpoints, with GRPO and DPO showing the most promising balance of performance and reliability.

| Method | AMD | Cataract | Grayscale | Tunnel Vision | Detached Retina | Avg. |
|---|---|---|---|---|---|---|
| ZS | 89.0 | 75.0 | 79.0 | 85.0 | 87.0 | 83.0 |
| KTO | 92.0 / 92.0 (+3.0 / +3.0) | 82.0 / 74.0 (+7.0 / -1.0) | 75.0 / 75.0 (-4.0 / -4.0) | 89.0 / 89.0 (+4.0 / +4.0) | 90.0 / 90.0 (+3.0 / +3.0) | 85.6 / 84.0 (+2.6 / +1.0) |
| GRPO | **95.0 / 95.0** (+6.0 / +6.0) | 79.0 / 78.0 (+4.0 / +3.0) | 83.0 / 83.0 (+4.0 / +4.0) | 91.0 / 91.0 (+6.0 / +6.0) | 90.0 / 90.0 (+3.0 / +3.0) | 87.6 / 87.4 (+4.6 / +4.4) |
| DPO | 94.0 / 94.0 (+5.0 / +5.0) | 82.0 / 76.0 (+7.0 / +1.0) | **84.0** / 81.0 (+5.0 / +2.0) | **92.0** / 90.0 (+7.0 / +5.0) | 88.0 / 88.0 (+1.0 / +1.0) | **88.0** / 85.8 (+5.0 / +2.8) |
| SFT | **95.0 / 95.0** (+6.0 / +6.0) | **84.0** / 75.0 (+9.0 / +0.0) | 79.0 / 79.0 (+0.0 / +0.0) | 85.0 / 85.0 (+0.0 / +0.0) | **92.0** / 92.0 (+5.0 / +5.0) | 87.0 / 85.2 (+4.0 / +2.2) |

Table 8: Adaptation performance under different visual distortions for **Imagenet** (Deng et al., 2009). We use Qwen2.5-VL 7B as speaker and listener. For each method we report two test accuracies: Maximum Test Accuracy / Test Accuracy of validation-set–picked checkpoint, and gains relative to **ZS**. Positive gains are in green, negatives in red. Best numbers per column are in **bold**. The last column reports average accuracies across distortions for both metrics, with gains relative to the ZS average.

| Method | AMD | Cataract | Grayscale | Tunnel Vision | Detached Retina | Avg. |
|---|---|---|---|---|---|---|
| ZS | 76.0 | 73.0 | 36.0 | 77.0 | 82.0 | 68.8 |
| KTO | 79.0/75.0 (+3.0/-1.0) | 75.0/72.0 (+2.0/-1.0) | **78.0**/73.0 (+42.0/+37.0) | 78.0/72.0 (+1.0/-5.0) | 84.0/84.0 (+2.0/+2.0) | 78.8/75.2 (+10.0/+6.4) |
| GRPO | 77.0/75.0 (+1.0/-1.0) | 73.0/73.0 (+0.0/+0.0) | 49.0/49.0 (+13.0/+13.0) | 79.0/76.0 (+2.0/-1.0) | 85.0/85.0 (+3.0/+3.0) | 72.6/71.6 (+3.8/+2.8) |
| DPO | 81.0/76.0 (+5.0/+0.0) | **77.0**/75.0 (+4.0/+2.0) | 75.0/40.0 (+39.0/+4.0) | 79.0/72.0 (+2.0/-5.0) | **86.0**/85.0 (+4.0/+3.0) | **79.6**/69.6 (+10.8/+0.8) |
| SFT | **91.0**/76.0 (+15.0/+0.0) | 73.0/62.0 (+0.0/-11.0) | 37.0/37.0 (+1.0/+1.0) | **88.0**/78.0 (+11.0/+1.0) | 85.0/85.0 (+3.0/+3.0) | 74.8/67.6 (+6.0/-1.2) |

Table 9: Adaptation performance under different visual distortions for **CLEVR** (Johnson et al., 2017). We use Qwen2.5-VL 7B as speaker and listener. For each method we report two test accuracies: Maximum Test Accuracy / Test Accuracy of validation-set–picked checkpoint, with gains relative to **ZS**. Positive gains are in green, negatives in red. Best numbers per row are in **bold**. The last column reports average accuracies across distortions for both metrics, with gains relative to the ZS average.

## B   ABLATION HETEROGENOUS MODEL SIZES

In this section, we conduct an ablation study to analyze the impact of using heterogeneous model sizes for the speaker and listener roles on CUB dataset. We explore four configurations using the Qwen-2.5 VL 3B and 7B models: a symmetric setup with matched sizes (3B/3B and 7B/7B) and asymmetric setups with mismatched sizes (3B speaker/7B listener and 7B speaker/3B listener). The results, presented in Tab. 11, evaluate the zero-shot (Base) performance and the maximum accuracy (Max) achieved after adaptation with GRPO and KTO.

Our first key observation relates to the zero-shot baseline performance. Counter-intuitively, the largest model configuration (7B/7B) does not yield the best baseline accuracy. In fact, it has the lowest average baseline among all setups (81.4%), primarily due to significant performance drops on the Cataract (63.0%) and Detached Retina (76.0%) distortions. The strongest baseline performance is achieved by the asymmetric configuration with a smaller speaker and a larger listener (3B/7B), which reaches an average accuracy of 88.8% for GRPO and 88.0% for KTO. This suggests that a more powerful listener model is crucial for robustly interpreting descriptions, even those generated by a smaller speaker.

Next, we analyze the gains from adaptation. We observe that both GRPO and KTO deliver consistent performance improvements across all model configurations. The magnitude of this improvement is most pronounced in the 7B/7B setup, which had the lowest baseline. For instance, GRPO and KTO improve the average accuracy by +6.2 and +6.4 points, respectively, with remarkable gains on the Cataract distortion (+19.0 for GRPO, +18.0 for KTO) where the baseline was weakest. This indicates that adaptation is highly effective at correcting the initial vulnerabilities of larger models.

From a practical standpoint, the optimal configuration for achieving the highest final accuracy is the 3B speaker and 7B listener setup. After adaptation with GRPO, this configuration reaches an average accuracy of 92.0%, the highest across all experiments. This result highlights a compelling strategy for efficient system

| Method | AMD | Cataract | Grayscale | Tunnel Vision | Detached Retina | Avg. |
|--------|-----|----------|-----------|---------------|-----------------|------|
| ZS | 90.0 | 63.0 | 76.0 | 90.0 | 88.0 | 81.4 |
| KTO | **94.0**/92.0 (+4.0/+2.0) | 81.0/76.0 (+18.0/+13.0) | 79.0/79.0 (+3.0/+3.0) | 92.0/92.0 (+2.0/+2.0) | 93.0/93.0 (+5.0/+5.0) | 87.8/86.4 (+6.4/+5.0) |
| GRPO | 93.0/91.0 (+3.0/+1.0) | 82.0/78.0 (+19.0/+15.0) | 81.0/77.0 (+5.0/+1.0) | 93.0/91.0 (+3.0/+1.0) | 93.0/93.0 (+5.0/+5.0) | **88.4**/86.0 (+7.0/+4.6) |
| DPO | 89.0/80.0 (-1.0/-10.0) | **89.0**/73.0 (+26.0/+10.0) | **85.0**/77.0 (+9.0/+1.0) | **96.0/96.0** (+6.0/+6.0) | 92.0/90.0 (+4.0/+2.0) | 90.2/83.2 (+8.8/+1.8) |
| SFT | 85.0/78.0 (-5.0/-12.0) | 77.0/74.0 (+14.0/+11.0) | 81.0/80.0 (+5.0/+4.0) | **96.0/96.0** (+6.0/+6.0) | **96.0/96.0** (+8.0/+8.0) | 87.0/84.8 (+5.6/+3.4) |

Table 10: Adaptation performance under different visual distortions for **CUB** (Wah et al., 2011). We use Qwen2.5-VL 7B as speaker and listener. For each method we report two test accuracies Maximum Test Accuracy / Test Accuracy of validation-set–picked checkpoint and gains relative to **ZS**. Positive gains are in green, negatives in red. Best numbers per row are in **bold**. The last column reports average accuracies across distortions for both metrics, with gains relative to the ZS average.

| Method | Model Configuration | | Type | Accuracy [%] | | | | |
|--------|---------|----------|------|-----|----------|-----------------|-----------|---------|
| | Speaker | Listener | | AMD | Cataract | Detached Retina | Grayscale | Average |
| **GRPO** | Qwen-2.5 VL 3B | Qwen-2.5 VL 3B | Base | 89.0 | 67.0 | 98.0 | 94.0 | 87.0 |
| | | | Max | 91.0 (+2.0) | 67.0 (+0.0) | 99.0 (+1.0) | 95.0 (+1.0) | 88.0 (+1.0) |
| | Qwen-2.5 VL 3B | Qwen-2.5 VL 7B | Base | 94.0 | 79.0 | 98.0 | 84.0 | 88.8 |
| | | | Max | 98.0 (+4.0) | 86.0 (+7.0) | 100.0 (+2.0) | 84.0 (+0.0) | 92.0 (+3.2) |
| | Qwen-2.5 VL 7B | Qwen-2.5 VL 3B | Base | 92.0 | 64.0 | 97.0 | 93.0 | 86.5 |
| | | | Max | 97.0 (+5.0) | 72.0 (+8.0) | 98.0 (+1.0) | 98.0 (+5.0) | 91.3 (+4.8) |
| | Qwen-2.5 VL 7B | Qwen-2.5 VL 7B | Base | 90.0 | 63.0 | 76.0 | 88.0 | 81.4 |
| | | | Max | 93.0 (+3.0) | 82.0 (+19.0) | 77.0 (+1.0) | 93.0 (+5.0) | 87.6 (+6.2) |
| **KTO** | Qwen-2.5 VL 3B | Qwen-2.5 VL 3B | Base | 88.0 | 67.0 | 96.0 | 94.0 | 86.3 |
| | | | Max | 90.0 (+2.0) | 70.0 (+3.0) | 100.0 (+4.0) | 98.0 (+4.0) | 89.5 (+3.2) |
| | Qwen-2.5 VL 3B | Qwen-2.5 VL 7B | Base | 94.0 | 80.0 | 98.0 | 80.0 | 88.0 |
| | | | Max | 96.0 (+2.0) | 85.0 (+5.0) | 99.0 (+1.0) | 82.0 (+2.0) | 90.5 (+2.5) |
| | Qwen-2.5 VL 7B | Qwen-2.5 VL 3B | Base | 92.0 | 67.0 | 96.0 | 93.0 | 87.0 |
| | | | Max | 93.0 (+1.0) | 75.0 (+8.0) | 97.0 (+1.0) | 97.0 (+4.0) | 90.8 (+3.8) |
| | Qwen-2.5 VL 7B | Qwen-2.5 VL 7B | Base | 90.0 | 63.0 | 76.0 | 88.0 | 81.4 |
| | | | Max | 94.0 (+4.0) | 81.0 (+18.0) | 79.0 (+3.0) | 93.0 (+5.0) | 87.8 (+6.4) |

Table 11: Detailed comparison of Zero-shot (Base) and Maximum Evaluation Accuracy across different distortions, including an overall Average on CUB dataset. Each model configuration is presented with separate rows for Base and Max accuracy, with gains over base shown in green.

design: pairing a smaller, computationally cheaper speaker model with a larger, more capable listener model provides a strong initial baseline that can be further enhanced through adaptation to achieve state-of-the-art performance. This asymmetric approach appears to offer a better trade-off between performance and computational resources than simply using the largest available models for both roles.

## C  PERFORMACE OF TOP TIER MODELS

To contextualize the performance of our adaptation methods, we conducted a comprehensive evaluation of several leading proprietary, closed-source large multimodal models (LMMs). We benchmarked their zero-shot performance on the ImageNet and CLEVR datasets under three challenging visual distortions: Cataract, Grayscale, and Tunnel Vision. The results, presented in Table 12 and Table 13, offer critical insights into the capabilities and limitations of current state-of-the-art models and highlight the value of targeted adaptation. We use the same prompting strategy as shown in main manuscript.

On the ImageNet dataset (Table 12), the top-tier models demonstrate remarkable robustness. Models like GPT-4o, GPT-5.1, and Claude-Sonnet-4 maintain exceptionally high accuracy, often exceeding 95% even with significant visual distortions. GPT-4o, for instance, achieves a perfect 100% accuracy on grayscale images, suggesting its internal representations are largely invariant to the absence of color information for this real-world object recognition task.

In comparison, the open-source Qwen2.5-VL 7B model in a zero-shot (ZS) setting shows a noticeable performance gap, with accuracies of 75.0% on Cataract, 79.0% on Grayscale, and 85.0% on Tunnel Vision. While these are respectable scores, they are clearly surpassed by the larger proprietary models. After fine-tuning with Direct Preference Optimization (DPO), the Qwen model's performance improves to 82.0%, 84.0%, and 92.0% respectively. This adaptation narrows the gap but does not close it entirely. The results on ImageNet suggest that for general perceptual tasks, the sheer scale and extensive pre-training of top-tier models provide a significant and durable advantage in robustness.

The results on the CLEVR dataset (Table 13) paint a dramatically different picture and underscore a critical vulnerability in large, generalist models. CLEVR requires compositional reasoning, a task that is more abstract than simple object recognition. While most top-tier models perform well under the Cataract and Tunnel Vision distortions, their performance collapses on grayscale images. For example, GPT-4o's accuracy plummets to 22.0%, and Gemini-3-Pro's drops to a mere 14.0%. The best-performing model, GPT-5.1, only reaches 36.0% accuracy, which is identical to the zero-shot performance of the much smaller Qwen2.5-VL 7B model. This indicates that the reliance on color cues for object differentiation and relational reasoning is a systemic weakness for un-adapted models on this synthetic dataset.

This is where the power of targeted adaptation becomes strikingly evident. After fine-tuning with DPO, the Qwen2.5-VL 7B model's accuracy on grayscale CLEVR images skyrockets from 36.0% to 75.0%. With this adaptation, the open-source model not only recovers its reasoning ability but vastly outperforms every single top-tier proprietary model on this specific task. Furthermore, its adapted score of 77.0% on Cataract is also superior to that of GPT-4o (74.0%) and Claude-Sonnet-4 (58.0%).

In summary, while top-tier models excel in general robustness on real-world image datasets like ImageNet, they can be surprisingly brittle when faced with specific distortions on tasks requiring complex reasoning, such as CLEVR. Our findings demonstrate that a smaller, open-source model can be fine-tuned to surpass these leading models in such challenging, niche scenarios, highlighting that targeted adaptation is a powerful and efficient strategy for achieving specialized robustness. Lastly, these results once again shed the light on the importance of task specific approaches for adaptation. Every task and distortion pair provides unique challenges. Some may be more distant from the learnt knowledge of the model and others may be closer. Lastly, we observe that GPT 5.1 shows the best trade off between the two datasets by showing average performance of 96.3% on Imagenet and taking a second place in CELVR dataset with 70% average accuracy.

## D  ABLATION OVER PROMPT TEMPLATE

Our standard prompt template , designed to elicit distinguishing features of the target image, is as follows:

```
You see two images.  Image 1:  [IMG_1], Image 2:  [IMG_2].
Focus on Image 1.  Your goal is to describe Image 1 in a
way that clearly distinguishes it from Image 2.  Do not
mention Image 2 in your description at all and just talk
about Image 1.  Highlight unique features of Image 1 as a
whole that differentiate it from Image 2.
```

| Speaker | Accuracy [%] | | | Avg |
|---|---|---|---|---|
| | Cataract | Grayscale | Tunnel Vision | |
| GPT-5.1 | **95.0** | 97.0 | 97.0 | **96.3** |
| GPT-4o | 94.0 | **100.0** | **99.0** | 97.7 |
| Claude-Sonnet-4 | **95.0** | 98.0 | **99.0** | 97.3 |
| Gemini-3-Pro | 92.0 | 98.0 | 96.0 | 95.3 |
| Grok-4 | 90.0 | 97.0 | 98.0 | 95.0 |
| Qwen2.5-VL 7B (ZS) | 73.0 | 36.0 | 77.0 | 62.0 |
| Qwen2.5-VL 7B (DPO) | 77.0 | 75.0 | 79.0 | 77.0 |

Table 12: Performance of top-tier speaker models on the **ImageNet** dataset, compared with an adapted open-source model (Qwen2.5-VL 7B). Listener is Qwen2.5-VL 7B.

| Speaker | Accuracy [%] | | | Avg |
|---|---|---|---|---|
| | Cataract | Grayscale | Tunnel Vision | |
| GPT-5.1 | 76.0 | 36.0 | **98.0** | 70.0 |
| GPT-4o | 74.0 | 22.0 | 88.0 | 61.3 |
| Claude-Sonnet-4 | 58.0 | 26.0 | 94.0 | 59.3 |
| Gemini-3-Pro | 61.0 | 14.0 | 89.0 | 54.7 |
| Grok-4 | 59.0 | 13.0 | 89.0 | 53.7 |
| Qwen2.5-VL 7B (ZS) | 73.0 | 36.0 | 77.0 | 62.0 |
| Qwen2.5-VL 7B (DPO) | **77.0** | **75.0** | 79.0 | **77.0** |

Table 13: Performance of top-tier speaker models on the **CLEVR** dataset, compared with an adapted open-source model. Note the dramatic improvement of the adapted model on grayscale images. Listener is Qwen2.5-VL 7B.

Table 14: Comparison of prompted vs. normal (Zero-Shot) accuracy for Grayscale and Cataract distortions on the CUB and CLEVR datasets. The "Prompted Acc." reflects performance with the distortion-aware instructions. We report results with Qwen2.5-VL 7B as listener and speaker both.

| Accuracy Type | Grayscale | Cataract |
|---|---|---|
| **Prompted Acc. [%]** | 74.0 | **80.0** |
| **Normal Acc. [%]** | **76.0** | 63.0 |

(a) Performance on the **CUB** dataset.

| Accuracy Type | Grayscale | Cataract |
|---|---|---|
| **Prompted Acc. [%]** | **50.0** | 56.0 |
| **Normal Acc. [%]** | 36.0 | **73.0** |

(b) Performance on the **CLEVR** dataset.

To make the model "distortion-aware," we introduce a listener-centered perspective by inserting an additional line of text just before the final sentence. This extra instruction explicitly describes the nature of the perceptual impairment. For the distortions analyzed here, the specific additions are:

- **For Grayscale:** `Your partner is sufffering from color blindness and will see a grayscale version of your image.`
- **For Cataract:** `Your partner is suffering from cataracts and will see a blurred version of your image along with some black artifacts.`

This modification directly tasks the model with generating descriptions that are robust and interpretable under the specified visual degradation. We report results with Qwen2.5-VL 7B as listener and speaker both.

The effectiveness of this distortion-aware prompting strategy varies significantly depending on the dataset and the nature of the distortion, as shown in Table 14.

On the **CUB dataset**, which features fine-grained classification of birds, the results are mixed. For the **Cataract**, distortion aware-prompting yields a substantial performance gain, increasing accuracy from 63.0% to 80.0%. This suggests that informing the model about the blur and artifacts successfully guides it to focus on more robust features. Conversely, for the **Grayscale** distortion, the prompted accuracy (74.0%) is slightly lower than the normal accuracy (76.0%). In the context of CUB, where bird species are often distinguished by subtle shapes, crests, and beak forms, color may be a less critical feature. The baseline model likely already focuses on these non-color features, and the explicit instruction about color blindness might act as a minor, unhelpful distraction, causing a negligible drop in performance.

On the **CLEVR dataset**, which consists of simple 3D shapes where color is a primary attribute, the impact of distortion aware-prompting is more dramatic and polarized. For the **Grayscale** distortion, the performance without the special prompt is extremely low (36.0%), as the loss of color information is catastrophic for a task

| Cataract-aware (distorted prompt) | Original (no distortion-aware prompt) |
|---|---|
| **sample 0** 
 The image showcases a variety of geometric shapes without mentioning layout or relative complexity. | Image 1 features a diverse collection of shapes with clear differences in layout, density, and object types (gold sphere, gray sphere, green cube), providing strong distinguishing cues. |
| **sample 1** 
 A collection of shapes with distinct colors and materials. A addCriterion (artifact) The image contains a green cylinder, purple cube, teal cube, teal cylinder, and golden sphere. | Image 1 features a coherent set of objects (green cylinder, purple cube, teal cube, teal cylinder, golden sphere) arranged cleanly on a gray background. |
| **sample 2** 
 A scene with a yellow sphere, gray sphere, and golden cylinder. addCriterion … | A clean description of three distinct objects with correct materials and spacing. |
| **sample 3** 
 A variety of geometric shapes. Red spheres stand out out their their addCriterion … | A detailed structured description listing cubes, spheres, cylinders in specific colors (red, green, yellow, gray, teal) with correct spatial distinctions. |
| **sample 4** 
 A simple arrangement of geometric shapes on a plain background. | Explicit listing of objects: green sphere, green cylinder, red cylinder, with explanation of spacing and composition. |
| **sample 6** 
 A variety of shapes including a gray cylinder and red cylinder; "shiny blue blue" addCriterion … | A precise object set: large gray cylinder, smaller red cylinder, shiny blue cube, two teal spheres, with a correct comparison of structure and density. |

Table 15: Representative CLEVR descriptions with and without cataract-aware prompting. Red = hallucinations, Blue = missing specificity, Purple = decoding artifacts. These error types explain why cataract-aware prompting degrades performance.

that heavily relies on it. By explicitly telling the model that its partner is colorblind, the prompted accuracy jumps to 50.0%. This significant improvement indicates that the prompt successfully forces the model to pivot its descriptive strategy away from the color information and towards alternative distinguishing features. In a striking reversal, for the **Cataract** distortion, the prompted accuracy plummets to 56.0% from a high normal accuracy of 73.0%. The baseline model, despite the blur. However, when prompted about cataracts, the model's performance collapses. This suggests a form of "over-correction," where the model, anticipating severe information loss, may generate overly simplistic or vague descriptions that fail to distinguish the target, or it may struggle to identify any feature it deems robust enough to mention.

We further examine the generations ot understand why cataract distortion aware prompt fails. We share some sample generations in Table. 15

To better understand the drop in accuracy for the Cataract condition on CLEVR, we qualitatively compare generations produced with and without cataract-aware prompting (Table 15). Three consistent patterns emerge.

**(1) Decoding instability.** Cataract-aware descriptions frequently include leaked internal tokens or partial loops (e.g., `addCriterion`), often appearing mid-sentence and disrupting otherwise coherent text. These artifacts occur far less frequently in the normal setting, where generations remain mostly intact even when imperfect.

**(2) Loss of discriminative detail.** Many cataract-aware outputs collapse into vague summaries (e.g., "a simple arrangement of geometric shapes") instead of enumerating object types, counts, or spatial cues. CLEVR depends heavily on such combinatorial details, and their omission removes the very signals needed for disambiguation.

**(3) Hallucinated content.** Cataract-aware prompting increases scene-independent errors, such as duplicated colors ("blue blue"), extra spheres, or incorrect materials. These hallucinations are especially damaging in CLEVR, where identity depends on simple, atomic properties like color, shape, and size.

Together, these effects indicate that cataract-aware prompting pushes the speaker into an over-cautious, unstable generative mode: it avoids strong commitments, produces underspecified or contradictory descriptions, and becomes more prone to decoding failures. In contrast, the normal prompt elicits structured, object-level descriptions that remain robust even under simulated blur. This disparity explains why distortion-aware prompting *reduces* accuracy for Cataract on CLEVR, despite improving performance in other settings such as CUB.

## E    Perceptual Distortions

## F    Hyperparameters

Unless otherwise noted, all models are adapted from `Qwen2.5-VL-7B`. Qwen2.5-VL-7B is used as a listener and a speaker. Furthermore, we also report results using Intervl3.5 in section 5.2. For preference data generation we use Qwen2.5-VL 32B with context length of 2048 tokens.

**Dataset-level defaults.**

- **CUB:** batch size $= 3$, learning rate $= 5 \times 10^{-6}$, sampling temperature $= 1.0$.
- **CLEVR:** batch size $= 3$, learning rate $= 5 \times 10^{-6}$, sampling temperature $= 1.1$.

**Method-specific settings.**

- **GRPO (online):** number of generations per instance $G = 4$; micro-batch size $= 1$; sampling temperature follows the dataset default (Imagnet: 1.0, CUB: 1.0, CLEVR: 1.1).
- **DPO (offline):** sampling temperature $= 1.0$; context length $= 2048$.
- **SFT (offline):** sampling temperature $= 1.0$; context length $= 2048$.

## G    Adapting Algorithms

To investigate adaptation under divergent visual spaces, we study both *offline* and *online* post-training algorithms. Offline methods allow for cost-effective *ante-hoc* adaptation when it is known in advance that a user population or set of agents shares a specific perceptual limitation. In contrast, online methods adapt through direct interaction with divergent agents, capturing the need for *in situ* personalization. Below we describe each algorithm in relation to the preference dataset $\{(\mathbf{x}, (m^+, m^-))\}$ introduced in Section 3.3, where $\mathbf{x} = (x_{\text{tgt}}, x_{\text{conf}})$ is the undistorted target–confound pair, $m^+$ denotes the distortion-consistent response, and $m^-$ the distortion-inconsistent response. In all cases, adaptation is applied via LoRA modules for efficiency.

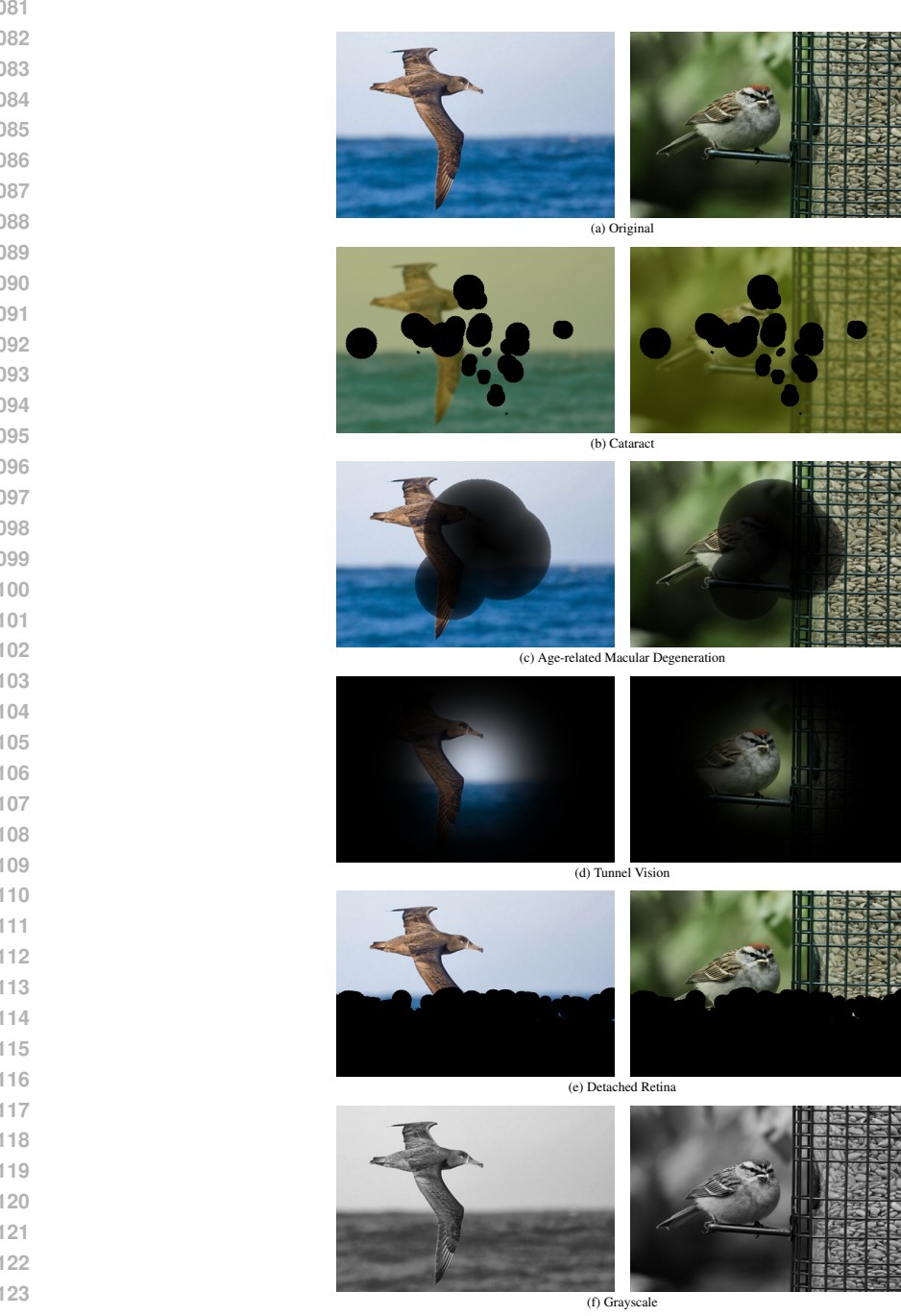

Figure 3: Examples of visual distortions used in our experiments. Each row shows two instances of the same impairment type, with the caption below.

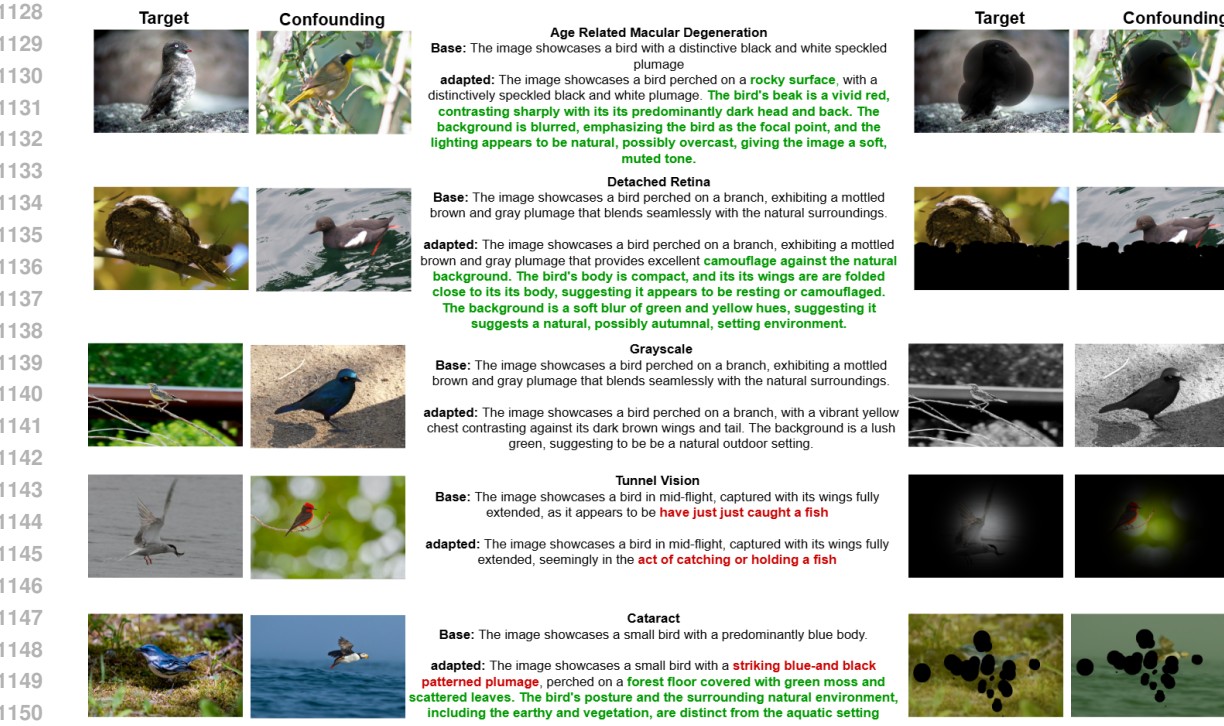

Figure 4: **Qualitative results For GRPO.** We share the qualitative results for GRPO (Shao et al., 2024) where left image is being described in each description. On left we share original images and on right we share distorted images. **Red text** represent change of information from original description while **bold text** represents new information.

**Supervised Fine-Tuning (SFT).** SFT provides a baseline for offline adaptation by directly fine-tuning the speaker on $(\mathbf{x}, m^+)$ pairs, ignoring the negative samples:

$$\mathcal{L}_{\text{SFT}}(\pi_\theta) = -\mathbb{E}_{(\mathbf{x},m^+)} \left[ \log \pi_\theta(m^+ \mid \mathbf{x}) \right]. \tag{1}$$

**Direct Preference Optimization (DPO).** DPO (Rafailov et al., 2023) directly optimizes the policy on preference pairs without a reward model. Given a reference model $\pi_{\text{ref}}$, the loss is

$$\mathcal{L}_{\text{DPO}}(\pi_\theta) = -\mathbb{E}_{(\mathbf{x},m^+,m^-)} \left[ \log \sigma \left( \beta \left( \log \frac{\pi_\theta(m^+|\mathbf{x})}{\pi_{\text{ref}}(m^+|\mathbf{x})} - \log \frac{\pi_\theta(m^-|\mathbf{x})}{\pi_{\text{ref}}(m^-|\mathbf{x})} \right) \right) \right], \tag{2}$$

where $\beta > 0$ controls the KL regularization.

**Kahneman–Tversky Optimization (KTO).** KTO (Ethayarajh et al., 2024) optimizes directly from preference pairs by modeling desirable responses $(m^+)$ and undesirable responses $(m^-)$ with asymmetric sensitivity. The implicit reward is

$$r_\theta(\mathbf{x}, m) = \log \frac{\pi_\theta(m \mid \mathbf{x})}{\pi_{\text{ref}}(m \mid \mathbf{x})}, \tag{3}$$

and the reference point is

$$z_0 = D_{\text{KL}}\big(\pi_\theta(\cdot \mid \mathbf{x}) \,\|\, \pi_{\text{ref}}(\cdot \mid \mathbf{x})\big). \tag{4}$$

The value function is then

$$v(\mathbf{x}, m) = \begin{cases} \lambda_+ \, \sigma\big(\beta(r_\theta(\mathbf{x}, m) - z_0)\big), & m = m^+, \\ \lambda_- \, \sigma\big(\beta(z_0 - r_\theta(\mathbf{x}, m))\big), & m = m^-, \end{cases} \tag{5}$$

where $\lambda_+, \lambda_- > 0$ control sensitivity to positive and negative samples, $\beta > 0$ adjusts curvature, and $\sigma$ is the sigmoid. The training objective is

$$\mathcal{L}_{\mathrm{KTO}}(\pi_\theta, \pi_{\mathrm{ref}}) \; = \; \mathbb{E}_{(\mathbf{x}, m^+, m^-) \sim D}\Big[\lambda_m - v(\mathbf{x}, m)\Big]. \tag{6}$$

This formulation captures prospect-theoretic asymmetry by treating distortion-consistent responses ($m^+$) and inconsistent responses ($m^-$) differently.

**Group Relative Policy Optimization (GRPO).** GRPO (Shao et al., 2024) is an online reinforcement learning algorithm that normalizes rewards across groups. For $\mathbf{x}$, we sample $\{m_i\}_{i=1}^G \sim \pi_{\theta_{\mathrm{old}}}(\cdot \mid \mathbf{x})$ with listener rewards $\{r_i\}$. The group-relative advantage is

$$A_i = \frac{r_i - \mathrm{mean}(\{r_j\}_{j=1}^G)}{\mathrm{std}(\{r_j\}_{j=1}^G)}. \tag{7}$$

The training objective is

$$J_{\mathrm{GRPO}}(\theta) = \mathbb{E}_{\mathbf{x}, \{m_i\}}\left[\frac{1}{G}\sum_{i=1}^G \min\left(\frac{\pi_\theta(m_i \mid \mathbf{x})}{\pi_{\theta_{\mathrm{old}}}(m_i \mid \mathbf{x})}A_i,\right.\right.$$
$$\left.\left.\mathrm{clip}\left(\frac{\pi_\theta(m_i \mid \mathbf{x})}{\pi_{\theta_{\mathrm{old}}}(m_i \mid \mathbf{x})}, 1 - \epsilon, \, 1 + \epsilon\right) A_i\right)\right] - \beta \, D_{\mathrm{KL}}(\pi_\theta \,\|\, \pi_{\mathrm{ref}}). \tag{8}$$

By leveraging relative comparisons within a group, GRPO stabilizes training while avoiding critic complexity.

## H  EXTENDED QUALITATIVE RESULTS

To better understand how adaptation manifests in practice, we provide qualitative results qualitative results for two best performing methods in online (KTO (Ethayarajh et al., 2024)) and offline methods (DPO (Rafailov et al., 2023)) in Fig. 5 and Fig. 5 respectively. In Fig. 5 and Fig. 6 the leftmost columns shows the original (target and confounding) images, while the rightmost columns shows the corresponding distorted versions. For more qualitative results, we refer the reader to the appendix H.

We show qualitative results for KTO (Ethayarajh et al., 2024) in Fig. (Ethayarajh et al., 2024). Each description is generated for the target image. Text in red indicates changes relative to the base description, and green text marks newly introduced information. Across distortions, we observe that speaker systematically adapts by enriching descriptions with context most relevant to the listener's perceptual limitations. For instance, under Age-Related Macular Degeneration (AMD), the adapted model supplements the base description with environmental cues such as the bird standing in shallow water, thereby leveraging contextual features when fine-grained details are obscured. Similarly, under Detached Retina, adaptation emphasizes the blurred green foliage in the background, aligning the description with the listener's altered visual field. In case of Grayscale, the adapted model highlights structural contrasts (e.g., "thin, bare branch"), compensating for the absence of color cues. In the case of Tunnel Vision, adaptation directs attention to sharp details of the bird's body and posture and emphasizes its presence in flight, providing information inside the narrow visual field of the listener. Finally, for Cataract, the adapted description incorporates references to the blurred green and brown background, again matching the listener's impaired visual input.

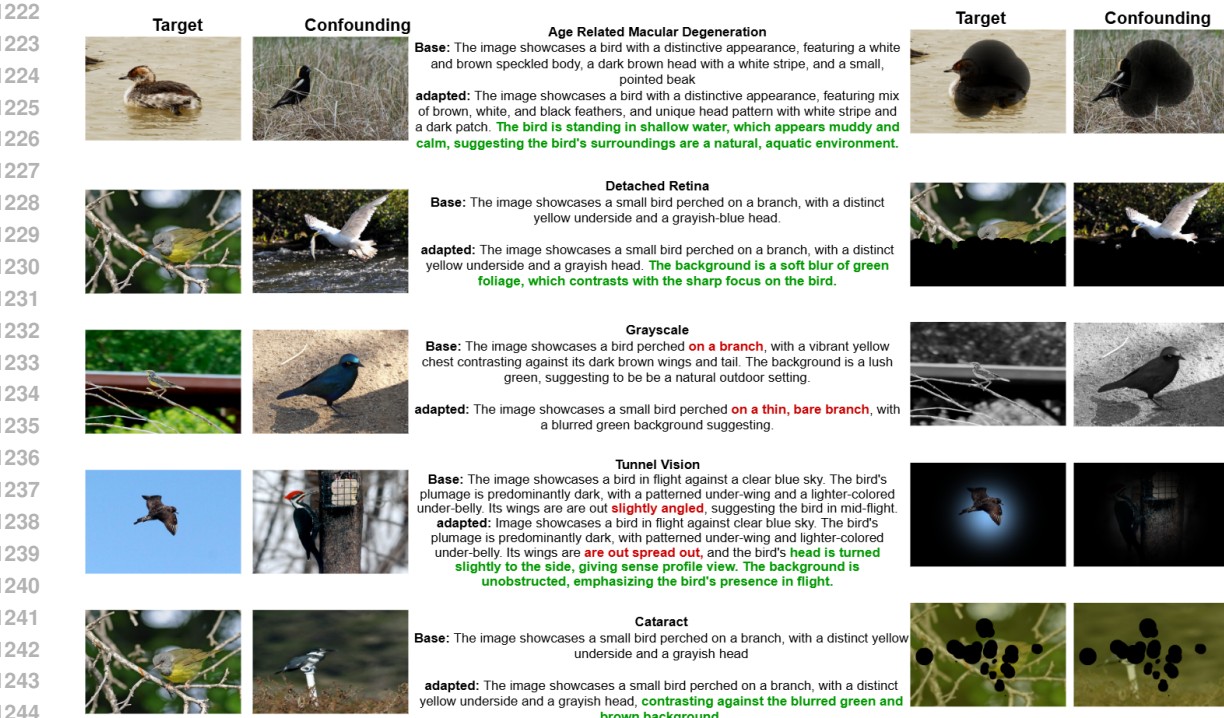

Figure 5: **Qualitative results For KTO (Ethayarajh et al., 2024)** with Qwen2.5-VL 7B as speaker and listener. The left image is being described in each description. On left we share original images and on right we share distorted images. In each description text with color blue represents the method name. **Red text** represent change of information from original description while **green text** represents new information.

We show qualitative results for DPO (Rafailov et al., 2023) (Rafailov et al., 2023). Each description is generated for the target image. Text in **red** indicates changes relative to the base description, and **green text** marks newly introduced information. Across distortions, we observe that DPO (Rafailov et al., 2023) enriches descriptions by emphasizing fine-grained attributes and structural cues that align with the listener's distorted perception. In case of Age-Related Macular Degeneration (AMD), for example, the adapted model introduces new details such as the "bird's beak being slightly open" and the broader "earthy-toned environment", compensating for the loss of central visual detail. For Detached Retina, adaptation emphasizes strong color contrasts (e.g., black head, orange-brown feathers) and background cues, allowing the description to remain informative despite missing peripheral vision. Additionally, emphasize on the upper part of the body of bird (e.g., black head) is also relevant to the listener because the body of the bird is occluded. In case of Grayscale, the adapted model highlights sharper contrasts in structure and position (e.g., "sharp focus on the bird against a softly blurred background" instead of blurred "green" background), directly addressing the absence of color cues. In the case of Tunnel Vision, descriptions adapt by focusing on body posture and motion, noting that the bird "dives towards the water" and has a "streamlined body", thereby ensuring that critical motion cues remain visible within the narrow field of vision. Additionally, we also notice that speaker changes 'bird just caught a fish' to bird diving towards water since the fish is not clearly visible in the narrow field of vision. Finally, under "Cataract", the adapted outputs include both fine visual traits (e.g., "short, pointed beak") and environmental grounding "natural, outdoor setting"), effectively compensating for the hazy appearance caused

by blurred input. These qualitative results show that DPO adaptation emphasizes discriminative features and contextual cues in ways that directly address distortion-specific limitations, complementing the quantitative findings where DPO often achieved the highest peak improvements. We also observe the level of details provided is more than KTO5 as the speaker is able to mention fine-grained details such as "bird's beak being slightly open". This further validates that in case of DPO6, speaker is benefittings strongly by distilling from a much stronger model with access to listener specific distortion.

Overall, these qualitative examples confirm that speaker can adapt not just by reformatting but by prioritizing perceptual attributes most relevant to listener in both online and offline settings.

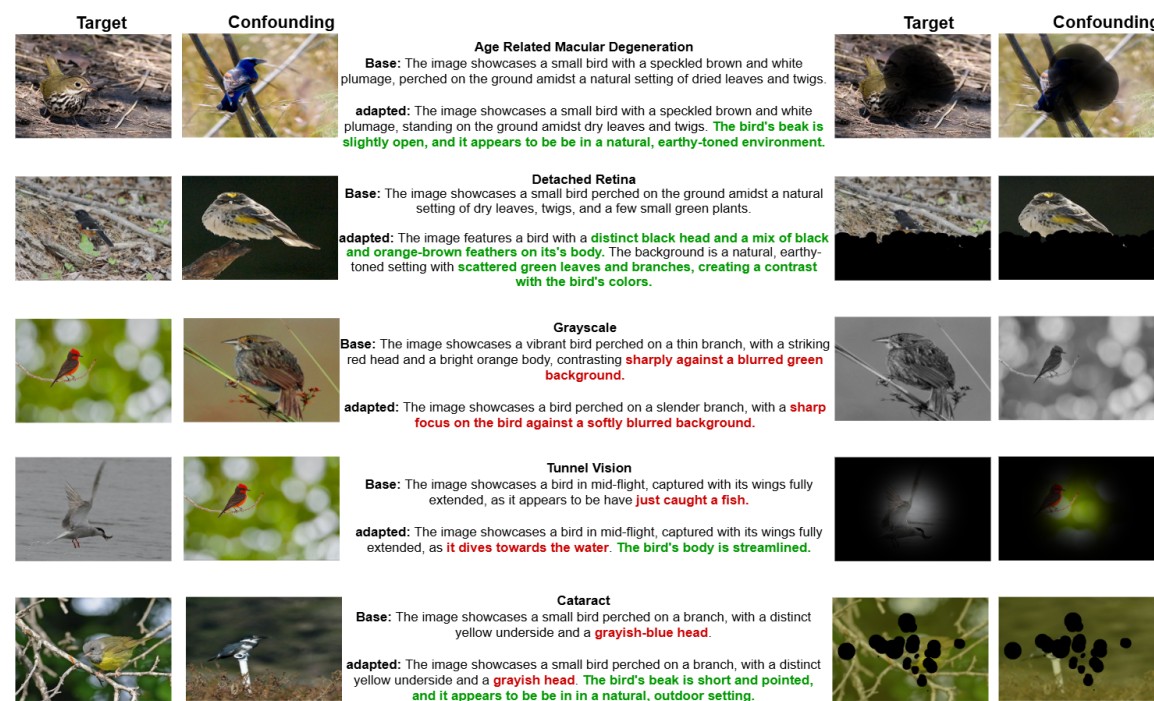

Figure 6: **Qualitative results For DPO** with Qwen2.5-VL 7B as speaker. The left image is being described in each description. On left we share original images and on right we share distorted images. In each description text with color blue represents the method name. **Red text** represent change of information from original description while **green text** represents new information.

## H.1 FAILURE CASES

In order to further validate our findings, We share failure cases in this section. We observe that in failure cases the adapted model produces descriptions that are misaligned with the listener's view (Fig. 7). In case of grayscale, the adapted model continues to emphasize color attributes under, such as describing "vibrant yellow plumage" or "subtle hints of green on the wings," while the listener cannot access color information. Another observation is the loss of contextual grounding, where the adapted description becomes overly concise and omits background cues or environmental context that could support disambiguation, especially under cataract

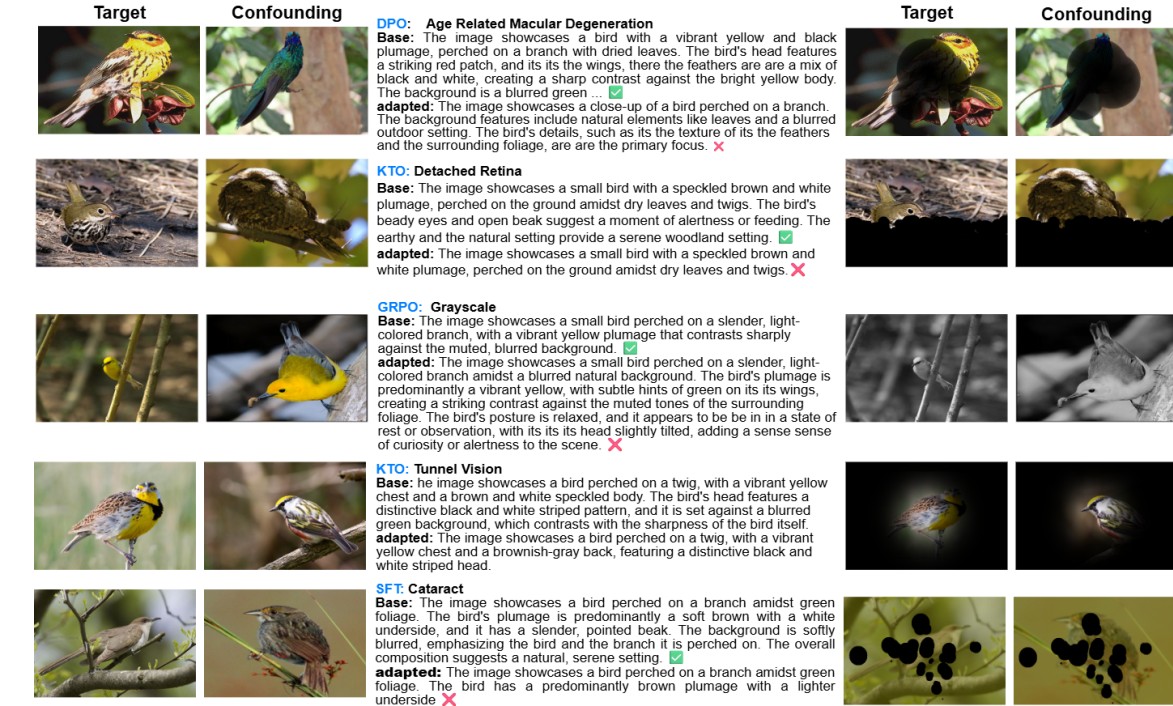

Figure 7: **Failure Cases** with Qwen2.5-VL 7B as speaker and listener. The left image is being described in each description. On left we share original images and on right we share distorted images. In each description tick represents correct and cross represents incorrect description. These are all the cases where adapted model fails and base model is successful.

or detached retina distortions. These errors reduce the interpretability and usefulness of adapted outputs, as the speaker either relies on inaccessible features or provides insufficient detail.

## I  DATASETS

To evaluate adaptation under divergent visual perceptions, we conduct experiments on two established multimodal benchmarks: **CLEVR** (Johnson et al., 2017) and **CUB** (Wah et al., 2011). These datasets offer complementary characteristics, enabling us to test both compositional reasoning and fine-grained visual recognition in our reference game setup.

**CLEVR. (Johnson et al., 2017)**   provides synthetic scenes with varied shapes, colors, materials. CLEVR contains 100k images, each depicting 3–10 objects rendered from a factorial combination of shapes, sizes, materials, and colors, with splits of 70k/15k/15k for training, validation, and testing.

**CUB. (Wah et al., 2011)**   provides natural, fine-grained imagery across 200 bird species. We form target–distractor pairs from visually similar birds to increase identification difficulty. The dataset includes

11,788 images, with roughly half used for training and half for testing, each annotated with bounding boxes, part keypoints, and over 300 binary attributes describing appearance and morphology.

## J    LINGUISTIC ANALYSIS

To better understand how adaptation methods alter the speaker's communication strategy, we conduct a linguistic analysis of the generated descriptions. This analysis reveals the specific ways in which adapted models shift their language to be more robust to the listener's perceptual limitations, providing quantitative evidence that complements our accuracy-based results.

a

### J.1    SHIFT IN TERM FREQUENCIES FOR GRAYSCALE DISTORTION

The most direct test of linguistic adaptation is to measure how the speaker's vocabulary changes when color information is removed.

Our analysis relies on a comprehensive taxonomy of visual attributes relevant to the CUB and CLEVR datasets. We constructed specific lexicons to detect chromatic colors (inaccessible under grayscale) versus achromatic terms and brightness descriptors (accessible). To capture the model's compensatory strategies, we tracked the frequency of alternative visual features, including shape (curved, pointed), texture (mottled, glossy), parts (crest, tail), pose (perched, soaring), and spatial relations.

Figure 8 presents a quantitative analysis of term category frequencies for the grayscale distortion, comparing the base model against models adapted with KTO and GRPO. The results show a clear and significant strategic shift.

As expected, both KTO and GRPO-adapted models dramatically reduce their reliance on color-based descriptors. The frequency of terms in the color and brightness categories decreases substantially, indicating that the models learn to avoid features that are inaccessible to the colorblind listener. More importantly, this reduction is accompanied by a compensatory increase in the use of other descriptive categories. We observe a considerable rise in the frequency of terms related to spatial arrangements, background context, and object parts. This demonstrates that the adapted speaker learns to ground its descriptions in structural and contextual cues—such as shape, position, and texture that remain invariant to the loss of color. This strategic pivot from inaccessible perceptual features to robust structural ones is a core mechanism behind the improved performance on all tasks as also illustrated in all the qualitative results.

### J.2    QUANTITATIVE LINGUISTIC SHIFTS ACROSS ALL DISTORTIONS

To complement our lexical analysis, we quantify how adaptation alters the readability and structural complexity of generated descriptions across distortions. We report two metrics—average word count and Flesch Reading Ease (FRE)—as shown in Fig. 9 (KTO) and Fig. 10 (GRPO). FRE is a standard measure of linguistic complexity defined as:

$$\text{FRE} = 206.835 - 1.015 \left( \frac{\text{total words}}{\text{total sentences}} \right) - 84.6 \left( \frac{\text{total syllables}}{\text{total words}} \right), \tag{9}$$

where higher scores indicate simpler text, and lower scores reflect denser or more complex phrasing.

Across distortions, we observe consistent and distortion-dependent shifts following adaptation:

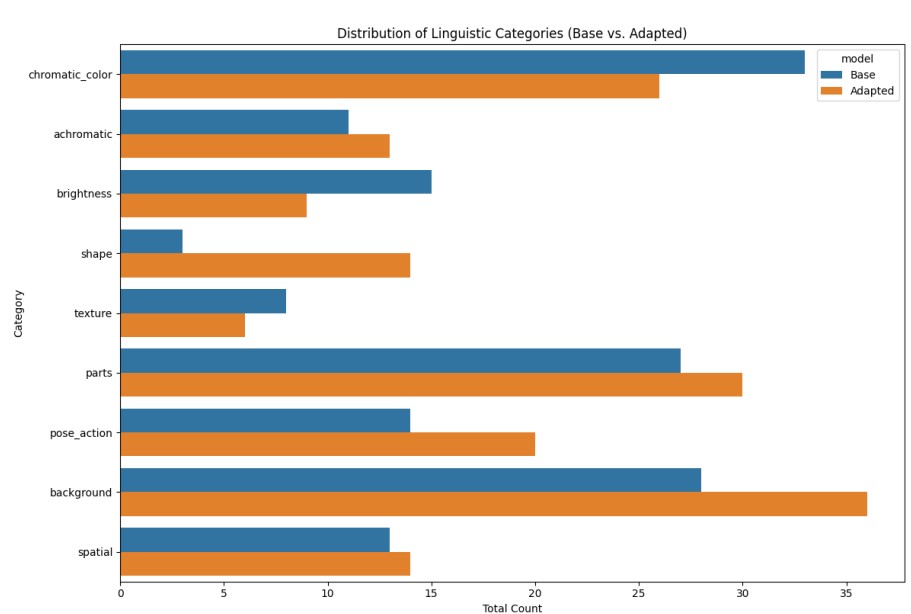

Figure 8: Term Category frequencies for grayscale distortion with KTO and GRPO both. We show here different category of terms used by our adapted model vs base model for grayscale images. We observe that the use of color terms and related terms such as brightness is less for our model whilespatial information such as background, spatial or parts are considerably higher in our adapted model.

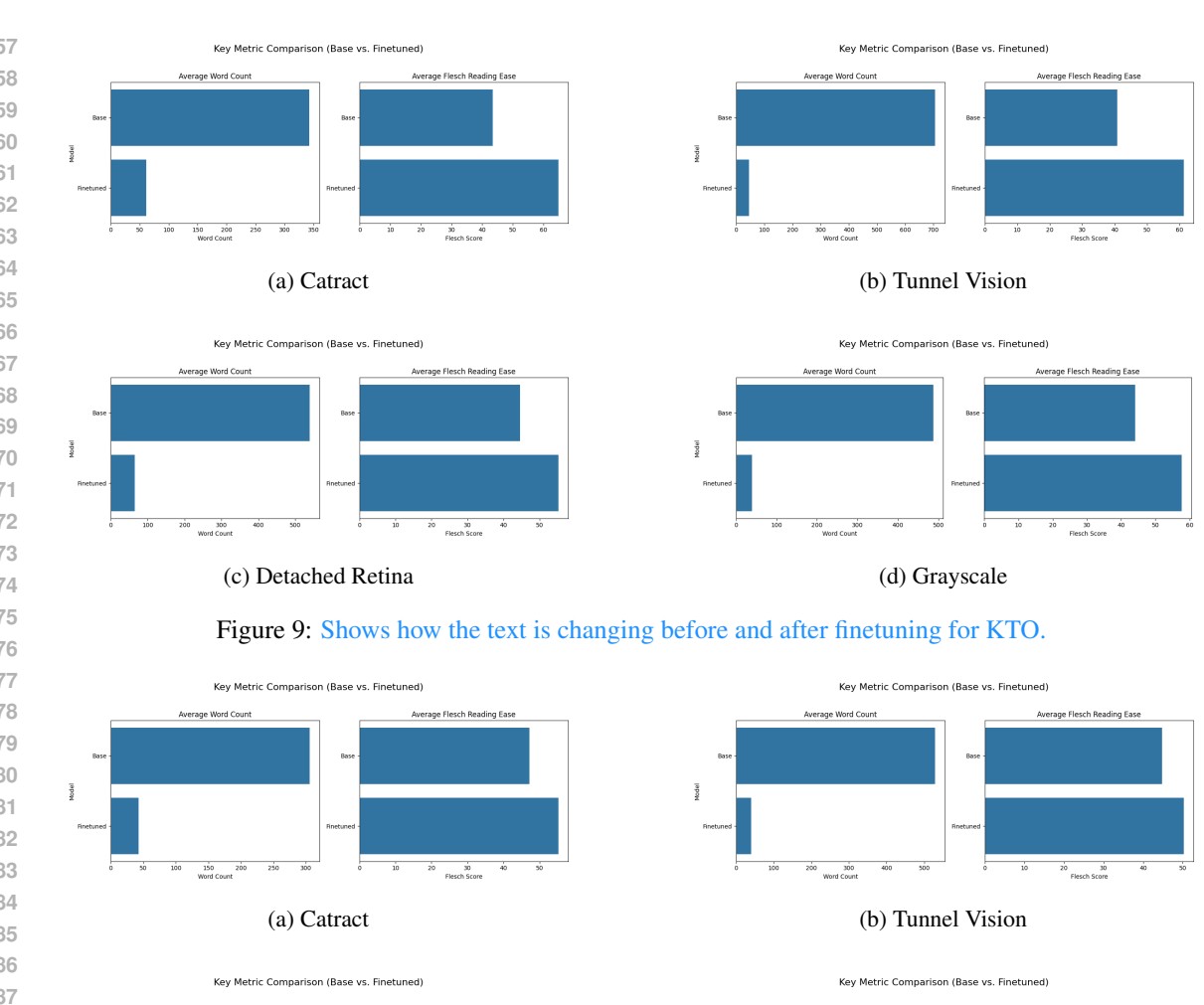

(a) Catract

(b) Tunnel Vision

(c) Detached Retina

(d) Grayscale

Figure 9: Shows how the text is changing before and after finetuning for KTO.

(a) Catract

(b) Tunnel Vision

(c) Detached Retina

(d) Grayscale

Figure 10: Shows how the text is changing before and after finetuning for GRPO.

**Overall Linguistic Shifts.** Across all distortion types, we observe a consistent pattern in both KTO and GRPO: the Flesch Reading Ease (FRE) scores increase while the average word count decreases after adaptation. This indicates that adapted speakers produce shorter and more readable descriptions, reflecting a systematic move toward clearer and more efficient communication. Furthermore, we observe higher flesch scores for KTO achieving around 60 score in 3/4 distortions. On the other hand, we observe slightly lower flesch scores in GRPO except in Grayscale. In all cases, we observe word count decreases after adaptation.

This shows that KTO aligns the model more robustly for communication with listener by making it more readable as compared to GRPO. Importantly, this shift suggests that adaptation encourages the model to rely on simpler, more direct phrasing that remains robust to perceptual impairments, rather than attempting to compensate with longer or more complex sentences.

## K   THEORETICAL ANALYSIS

The goal of this section is to provide a theoretical explanation for the empirical behaviors observed across our experiments on CLEVR and CUB. Our results reveal three consistent patterns in how GRPO, KTO, and DPO adapt speakers under divergent visual spaces: (i) GRPO yields modest but stable improvements but often plateaus. We show it can happen when reward becomes low-variance; (ii) KTO provides high peak gains. We show low variance (reward degeneracy) does not impact KTO; and (iii) DPO can achieve high peak performance offline with greedy sampling but exhibits instability with stochastic sampling. We show that it is because DPO learns to put all mass on the preferred trajectories.

To understand why these behaviors arise, we analyze the underlying gradient structures of GRPO, KTO, and DPO. We show:

- **GRPO stalls when reward variance is low.** GRPO normalizes rewards within each sampled group, and its update magnitude is proportional to the within-group reward variance. We prove that in cases where reward variance collapses to zero, GRPO's reward-driven gradient is exactly zero, explaining the empirical plateaus observed in our online CLEVR and CUB experiments.

- **KTO continues improving even under degenerate rewards.** Unlike GRPO, KTO does not depend on reward heterogeneity. Instead, its gradient is driven by log-likelihood ratios with respect to a fixed reference policy. We show that this gradient remains strictly nonzero even when rewards are constant or uninformative, mirroring the stable improvements KTO achieves across all distortions.

- **DPO pushes probability mass toward preferred responses.** For the two-trajectory case, we prove that DPO strictly increases the probability of the preferred response and is maximized in the limit where the model assigns all probability mass to the positive sample. This theoretical behavior aligns with our empirical findings: DPO achieves the strongest peak improvements offline with greedy sampling but may fail under stochastic sampling.

Taken together, these results provide a unified theoretical grounding for the empirical phenomena observed throughout the main paper: GRPO tends to be conservative but stable because of reward degeneracy leading to vanishing gradients, KTO shows high peak gains because it is robust to reward degeneracy, and DPO inherently encourages probability concentration around preferred trajectories, explaining its strong performance under greedy sampling.

### K.1 PRELIMINARIES

Fix an input $x$ and group size $G \geq 2$. For GRPO, we draw $m_i \sim \pi_{\theta_{\text{old}}}(\cdot \mid x)$ and obtain binary listener rewards $r_i \in \{0, 1\}$. Let

$$\bar{r} = \frac{1}{G} \sum_{i=1}^{G} r_i, \qquad s = \text{std}\big(\{r_i\}_{i=1}^{G}\big),$$

where $\mathrm{std}$ denotes either the sample standard deviation $s^2 = \frac{1}{G-1}\sum_i(r_i - \bar{r})^2$ ("unbiased") or the population standard deviation $s^2 = \frac{1}{G}\sum_i(r_i - \bar{r})^2$. Define the group-relative advantage

$$A_i = \frac{r_i - \bar{r}}{s},$$

with the standard implementation convention $A_i = 0$ if $s = 0$ (to avoid division by 0). Ignoring clipping and importance ratios (this only strengthens the conclusions; see Remark 1), the GRPO group term has policy-gradient proportional to

$$g_{\mathrm{GRPO}} = \frac{1}{G}\sum_{i=1}^{G} A_i \, \nabla_\theta \log \pi_\theta(m_i \mid x) \tag{10}$$

plus the separate $-\beta\nabla_\theta \mathrm{DKL}(\pi_\theta \| \pi_{\mathrm{ref}})$ regularizer.

For KTO, define

$$r_\theta(x,m) = \log\frac{\pi_\theta(m \mid x)}{\pi_{\mathrm{ref}}(m \mid x)}, \qquad z_0(x) = \mathrm{DKL}\big(\pi_\theta(\cdot \mid x) \, \| \, \pi_{\mathrm{ref}}(\cdot \mid x)\big),$$

and

$$v(x,m) = \begin{cases} \lambda_+ \, \sigma\big(\beta[r_\theta(x,m) - z_0(x)]\big), & m = m^+, \\ \lambda_- \, \sigma\big(\beta[z_0(x) - r_\theta(x,m)]\big), & m = m^-. \end{cases}$$

The training objective is $L_{\mathrm{KTO}} = \mathbb{E}[\lambda_m - v(x,m)]$.

For DPO, with pair $(m^+, m^-)$ at fixed $x$,

$$L_{\mathrm{DPO}}(\theta) = -\mathbb{E}\Big[\log\sigma\Big(\beta\Big\{\underbrace{\log\pi_\theta(m^+ \mid x) - \log\pi_\theta(m^- \mid x)}_{\Delta_\theta} - \underbrace{\log\pi_{\mathrm{ref}}(m^+ \mid x) + \log\pi_{\mathrm{ref}}(m^- \mid x)}_{\Delta_{\mathrm{ref}}}\Big\}\Big)\Big]. \tag{11}$$

## K.2 GRPO: UPDATE MAGNITUDE VANISHES WITH LOW REWARD VARIANCE

**Lemma 1** (Standardized residual identities). *On any non-degenerate group ($s > 0$), the standardized residuals $A_i = (r_i - \bar{r})/s$ satisfy*

$$\sum_{i=1}^{G} A_i = 0, \qquad \sum_{i=1}^{G} A_i^2 = \kappa_G,$$

*where*

$$\kappa_G = \begin{cases} G - 1, & \text{if } s^2 = \frac{1}{G-1}\sum_i(r_i - \bar{r})^2 \text{ (sample std)}, \\ G, & \text{if } s^2 = \frac{1}{G}\sum_i(r_i - \bar{r})^2 \text{ (population std)}. \end{cases}$$

*Proof.* $\sum_i A_i = \frac{1}{s}\sum_i(r_i - \bar{r}) = 0$ by definition of the mean. For the sum of squares, $\sum_i A_i^2 = \sum_i(r_i - \bar{r})^2/s^2$, which equals $G - 1$ or $G$ depending on the convention for $s^2$. $\square$

**Proposition 1** (Zero-variance groups stall the GRPO reward update). *If the group rewards are identical, $r_1 = \cdots = r_G$ (equivalently, $s = 0$), then by the $A_i = 0$ convention the group-gradient in equation 10 is exactly $0$. The only remaining update is the KL regularizer $-\beta \nabla_\theta \mathrm{DKL}(\pi_\theta \| \pi_{\mathrm{ref}})$.*

**Proposition 2** (Expected GRPO update is controlled by reward heterogeneity). *Assume $r_i \overset{\text{i.i.d.}}{\sim} \mathrm{Bernoulli}(p)$ (conditioned on $x$). Let $D$ be the event "the group is degenerate", i.e., all $r_i$ are equal. Then*

$$\mathbb{P}(D) = p^G + (1-p)^G, \qquad \mathbb{P}(D^c) = 1 - p^G - (1-p)^G.$$

*Define the constant $C_G := \sqrt{\kappa_G / G}$, with $\kappa_G$ as in Lemma 1.*

*(a) If there exists $B < \infty$ with $\|\nabla_\theta \log \pi_\theta(m \mid x)\| \leq B$ almost surely, then*

$$\mathbb{E}\big[\|g_{\mathrm{GRPO}}\|\big] \leq C_G\, B \left(1 - p^G - (1-p)^G\right). \tag{12}$$

*(b) Without a uniform bound, if $\mathbb{E}\big[\|\nabla_\theta \log \pi_\theta(m \mid x)\|^2\big] \leq V$ for some $V < \infty$, then by Cauchy–Schwarz*

$$\mathbb{E}\big[\|g_{\mathrm{GRPO}}\|\big] \leq C_G\, \sqrt{V}\, \sqrt{1 - p^G - (1-p)^G}. \tag{13}$$

*In either case the right-hand side $\to 0$ as the Bernoulli variance $p(1-p) \to 0$ (i.e., $p \to 0$ or $p \to 1$).*

*Proof.* On $D$ we have $g_{\mathrm{GRPO}} = 0$ by Prop. 1. On $D^c$, Lemma 1 and Cauchy–Schwarz give

$$\left\| \frac{1}{G} \sum_{i=1}^G A_i \nabla \log \pi_\theta(m_i \mid x) \right\| \leq \frac{1}{G} \sqrt{\sum_i A_i^2} \sqrt{\sum_i \|\nabla \log \pi_\theta(m_i \mid x)\|^2} \leq \frac{\sqrt{\kappa_G}}{\sqrt{G}} \cdot \begin{cases} B, & \text{(a)} \\ \sqrt{\sum_i \|\nabla \log \pi_\theta(m_i \mid x)\|^2}, & \text{(b)}. \end{cases}$$

(a) Take expectation and multiply by $\mathbb{P}(D^c)$.

(b) Using $\mathbb{E}\big[\sqrt{Y}\, \mathbf{1}_{D^c}\big] \leq \sqrt{\mathbb{E}[Y]}\, \sqrt{\mathbb{P}(D^c)}$ with $Y = \sum_i \|\nabla \log \pi_\theta(m_i \mid x)\|^2$ yields $\mathbb{E}\big[\sqrt{Y}\, \mathbf{1}_{D^c}\big] \leq \sqrt{GV}\, \sqrt{\mathbb{P}(D^c)}$, which gives equation 13. $\qquad \square$

**Corollary 1** (Low-variance $\Rightarrow$ frequent stalling). *For binary rewards and group size $G$, $\mathbb{P}(D) = p^G + (1-p)^G \to 1$ as $p(1-p) \to 0$. Thus, in low-variance regimes (e.g., near-deterministic listener feedback), GRPO's reward-driven update vanishes on most groups (up to the separate KL regularizer).*

*Remark* 1 (Clipping and importance ratios). GRPO uses importance ratios $\rho_i = \pi_\theta(m_i \mid x) / \pi_{\theta_{\mathrm{old}}}(m_i \mid x)$ and a clipped surrogate. Both the $\min(\cdot, \cdot)$ form and clipping factors can only reduce the magnitude of the per-group update relative to the un-clipped, un-min'ed expression used in equation 10. Hence the bounds above remain valid or become tighter when those stabilizers are present.

### K.3 KTO: GRADIENT DOES NOT STALL WITH DEGENERATE REWARDS

**Proposition 3** (KTO keeps training even when rewards are degenerate). *For $L_{\mathrm{KTO}} = \mathbb{E}[\lambda_m - v(x, m)]$,*

$$\nabla_\theta L_{\mathrm{KTO}} = -\mathbb{E}\Big[\lambda_+ \beta \sigma'\big(\beta[r_\theta - z_0]\big)\big(\nabla r_\theta - \nabla z_0\big)\mathbf{1}\{m = m^+\}\Big] \tag{14}$$

$$+ \mathbb{E}\Big[\lambda_- \beta \sigma'\big(\beta[z_0 - r_\theta]\big)\big(\nabla r_\theta - \nabla z_0\big)\mathbf{1}\{m = m^-\}\Big],$$

where $\nabla r_\theta(x, m) = \nabla \log \pi_\theta(m \mid x)$ and $\nabla z_0(x) = \nabla \mathrm{DKL}(\pi_\theta(\cdot \mid x) \| \pi_{\mathrm{ref}}(\cdot \mid x))$. In particular, at $\pi_\theta = \pi_{\mathrm{ref}}$ we have $r_\theta \equiv 0$, $z_0 \equiv 0$, $\sigma'(0) = \frac{1}{4}$, and $\nabla z_0 = 0$, hence

$$\nabla_\theta L_{\mathrm{KTO}}\Big|_{\pi_\theta = \pi_{\mathrm{ref}}} = -\frac{\beta}{4}\,\mathbb{E}\big[\lambda_+ \nabla \log \pi_\theta(m^+ \mid x) - \lambda_- \nabla \log \pi_\theta(m^- \mid x)\big], \tag{15}$$

which is generically nonzero. Therefore KTO continues to make progress regardless of reward degeneracy or low reward variance.

*Proof.* Differentiate $v(x, m)$ case-wise using the chain rule; $\sigma'(t) = \sigma(t)[1 - \sigma(t)] > 0$. For $m = m^+$, $v = \lambda_+ \sigma(\beta[r_\theta - z_0])$, hence $\nabla v = \lambda_+ \beta \sigma'(\beta[r_\theta - z_0])(\nabla r_\theta - \nabla z_0)$; for $m = m^-$, $v = \lambda_- \sigma(\beta[z_0 - r_\theta])$, hence $\nabla v = \lambda_- \beta \sigma'(\beta[z_0 - r_\theta])(\nabla z_0 - \nabla r_\theta) = -\lambda_- \beta \sigma'(\beta[z_0 - r_\theta])(\nabla r_\theta - \nabla z_0)$. Substituting into $\nabla L_{\mathrm{KTO}} = -\mathbb{E}[\nabla v]$ yields equation 14. At $\pi_\theta = \pi_{\mathrm{ref}}$, we have $\nabla z_0 = 0$ since $\mathrm{DKL}(\cdot \| \pi_{\mathrm{ref}})$ has a minimum at $\pi_{\mathrm{ref}}$ (alternatively, differentiate the definition of the KL and use $\sum_m \nabla \pi_\theta(m \mid x) = 0$). Plugging in gives equation 15. $\square$

*Remark* 2. Unlike GRPO, KTO does not normalize by group variance and does not require heterogeneous rewards; it drives learning via log-likelihood ratios to a fixed reference. Hence it does not stall when rewards are constant.

### K.4 DPO: MASS CONCENTRATION ON PREFERRED/HIGH-LIKELIHOOD TRAJECTORIES

**Theorem 1** (Two-trajectory case: DPO is maximized at $p \to 1$)**.** *Fix $x$ and a single pair $(m^+, m^-)$. Let $p = \pi_\theta(m^+ \mid x) \in (0, 1)$ so $\pi_\theta(m^- \mid x) = 1 - p$, and let $c = \Delta_{\mathrm{ref}} = \log \pi_{\mathrm{ref}}(m^+ \mid x) - \log \pi_{\mathrm{ref}}(m^- \mid x)$. Then*

$$-L_{\mathrm{DPO}}(p) = \log \sigma\Big(\beta\Big(\log \frac{p}{1-p} - c\Big)\Big),$$

*and*

$$\frac{d}{dp}\big[-L_{\mathrm{DPO}}(p)\big] = \beta\,\sigma'\Big(\beta\Big(\log \frac{p}{1-p} - c\Big)\Big)\frac{1}{p(1-p)} > 0 \quad \text{for } p \in (0, 1).$$

*Remark* 3. Thus $-L_{\mathrm{DPO}}(p)$ is strictly increasing in $p$ and is maximized at $p \to 1$: DPO concentrates mass on $m^+$.

### K.5 TAKEAWAYS

- GRPO's reward-driven update vanishes exactly on groups with zero within-group reward variance; with Bernoulli rewards the frequency of such groups $\to 1$ as $p(1 - p) \to 0$ (Cor. 1). Bounds equation 12 and equation 13 make the dependence explicit.
- KTO's gradient depends on log-likelihood ratios to a fixed reference and remains nonzero even with degenerate rewards; the corrected unified gradient is equation 14, and at $\pi_\theta = \pi_{\mathrm{ref}}$ it reduces to equation 15.
- DPO strictly increases the probability of the preferred response in the two-trajectory case (Thm. 1),