# OpenReview forum: "When We Don’t See the Same Picture: Aligning Agents with Divergent Visual Spaces"
_ICLR.cc/2026/Conference — Submitted to ICLR 2026_

### Official Review · Reviewer_zDhq · 2025-10-30

**Soundness:** 3
**Presentation:** 3
**Contribution:** 3
**Rating:** 6
**Confidence:** 4

**Summary:**

This paper addresses an important and timely problem: how AI agents can communicate effectively when they perceive the world differently. Using an image reference game with five perceptual distortions inspired by real human visual impairments, the authors compare offline (SFT, DPO) and online (KTO, GRPO) adaptation methods. The core motivation—enabling better human-AI collaboration under perceptual asymmetries—is compelling and has clear humanitarian implications for accessibility. The experimental framework is well-designed, and the preference dataset construction is creative.

**Strengths:**

- Novel and socially relevant problem formulation. The scenario—communication between agents with mismatched visual perception—is both conceptually interesting and practically important, especially for accessibility and assistive AI.

- Systematic experimental framework. The paper compares both offline and online post-training paradigms under a controlled setup, with extensive quantitative and qualitative analyses.

- Strong execution and clarity. The distortion modeling (cataract, AMD, color blindness, etc.) is well-motivated and grounded in real-world perceptual conditions.

- Potential for human-AI applications, such as blind assistance, human–robot collaboration, and adaptive visual reasoning.

**Weaknesses:**

- Limited conceptual depth in algorithmic comparison. The paper mainly benchmarks standard SFT/DPO/KTO/GRPO methods without offering new algorithmic contributions or deeper theoretical insight into why some alignments succeed or fail.

- Shallow treatment of RL-based adaptation. While four algorithms are compared, the discussion of their distinct behaviors or learning dynamics is limited.

- Cross-agent results (e.g., Qwen2.5-VL vs. InternVL3.5) show small or inconsistent gains but lack detailed error analysis.

**Questions:**

- I am very interested in how different top-tier multimodal LLMs (e.g., GPT-5, Gemini 2.5 Pro) would perform as speakers in this framework. Could the authors test or discuss such systems? This would greatly inform model choice for assistive applications.

- Have the authors considered deploying the speaker–listener setup in realistic blind-assistance contexts (e.g., identifying street objects, reading signs)? Even small-scale trials could strongly strengthen the paper’s societal impact.

- Could the authors explore human-in-the-loop experiments, where human participants act as listeners (receiving speech-based descriptions)? Given that humans have limited auditory memory span, such tests could reveal important differences between human and model listeners.

---

> ### Author Response · Authors · 2025-11-25
>
> We thank the reviewer for the thoughtful reviews. We address each concern step by step below.
>
>
> **1. I am very interested in how different top-tier multimodal LLMs (e.g., GPT-5, Gemini 2.5 Pro) would perform as speakers in this framework. Could the authors test or discuss such systems? This would greatly inform model choice for assistive applications.**
>
> > **Answer:** We share the results in section C of supplementary material.
> ### Performance on the ImageNet dataset
>
> | Speaker            | Cataract | Grayscale | Tunnel Vision |
> |--------------------|----------|-----------|----------------|
> | GPT-5.1            | 95.0     | 97.0      | 97.0           |
> | GPT-4o             | 94.0     | 100.0     | 99.0           |
> | Claude-Sonnet-4    | 95.0     | 98.0      | 99.0           |
> | Gemini-3-Pro       | 92.0     | 98.0      | 96.0           |
> | Grok-4             | 90.0     | 97.0      | 98.0           |
> | **Qwen2.5-VL 7B (ZS)**  | 73.0     | 36.0      | 77.0           |
> | **Qwen2.5-VL 7B (DPO)** | 77.0     | 75.0      | 79.0           |
>
> ### Performance on the CLEVR dataset
>
> | Speaker            | Cataract | Grayscale | Tunnel Vision |
> |--------------------|----------|-----------|----------------|
> | GPT-5.1            | 76.0     | 36.0      | 98.0           |
> | GPT-4o             | 74.0     | 22.0      | 88.0           |
> | Claude-Sonnet-4    | 58.0     | 26.0      | 94.0           |
> | Gemini-3-Pro       | 61.0     | 14.0      | 89.0           |
> | Grok-4             | 59.0     | 13.0      | 89.0           |
> | **Qwen2.5-VL 7B (ZS)**  | 73.0     | 36.0      | 77.0           |
> | **Qwen2.5-VL 7B (DPO)** | 77.0     | 75.0      | 79.0           |
>
>
> We observe that GPT-5.1 gives the best trade-off between CLEVR and CUB. On CUB, it is on par with other models but on CLEVR, it outperforms significantly. We also compare it with a simple Qwen 2.5-VL 7B speaker in Zero shot setting and after finetuning with DPO. Results show that on more difficult CLEVR dataset, finetuning outperforms top tier models while on Imagenet, it lags behind.
>
>
> **2. Have the authors considered deploying the speaker–listener setup in realistic blind-assistance contexts (e.g., identifying street objects, reading signs)? Even small-scale trials could strongly strengthen the paper’s societal impact. Can we have humans in the loop?**
>
> > **Answer:** Thank you for the suggestion. However, given the short rebuttal time, it is not possible to get access to perceptually challenged people to perform this user study rigorously.
>
> Furthermore, humans-in-the-loop training is equally challenging since the methods require hundreds of interactions. In a real-world deployment over a longer horizon, this is feasible, but we will have to leave a proper human study to future research at this time.  We note that ImageNet experiments (Table 8) demonstrate the framework already extends to naturalistic images, showing +4–5% gains even from a strong 83% baseline, suggesting feasibility for future real-world deployment.

---

> ### Author Response · Authors · 2025-11-25
>
> **3. Limited conceptual depth in algorithmic comparison. The paper mainly benchmarks standard SFT/DPO/KTO/GRPO methods without offering new algorithmic contributions or deeper theoretical insight into why some alignments succeed or fail.**
>
> > **Answer:** While our core contribution is a technical investigation of AI-to-AI communication under perceptual mismatch, the fundamental motivation for modeling divergent visual perception stems from the critical need for accessible and inclusive AI systems. An estimated 2.2 billion people live with vision impairment or blindness according to the World Health Organization (WHO) [2]. The sheer scale of this population underscores the necessity for LLMs to communicate effectively with users whose visual experiences differ significantly from the ``normal'' vision assumed by most models. Ensuring that LLMs can adapt their communication to users with divergent perceptual abilities is not merely a matter of engineering efficiency, but a prerequisite for deploying equitable, human-centric AI that serves the entire global population.
>
> **Shallow treatment of RL-based adaptation. While four algorithms are compared, the discussion of their distinct behaviors or learning dynamics is limited.**
> > **Answer:** We argue that DPO outperforms because of the high quality dataset. On the other hand, recent works have cited GRPO reward function is prone to gradient vanishing in low variance reward datasets [3] which can lead to under exploration. We believe that this is the reason behind KTO high peak performances and GRPO consistent and more uniform improvements. Our main purpose of study is to highlight the unique problem formulation of communicating agents with divergent visual spaces, process for alignment in such a problem setting and additionally, we highlight interesting observations such as KTO shows more robustness, offline adaptation is more effective even without access to the listener dynamics and SFT also shows strong robustness to distribution shifts.
>
>
>
> [1] Nwabueze, Josephine Nneka. "BARRIERS GLOBALLY FACED BY PERSONS WITH DISABILITIES." Special needs education from the lens of interdisciplinary dialogue: a festschrift in honour of prof. Emeka d. Ozoji 3 (2023).
>
> [2] World Health Organization and The Lancet Global Health Commission on Global Eye Health. World Report on Vision. World Health Organization, 2021. URL https://www.who.int/publications/i/ item/9789241517402.
>
> [3] Leng, Sicong, et al. "MMR1: Enhancing Multimodal Reasoning with Variance-Aware Sampling and Open Resources." arXiv preprint arXiv:2509.21268 (2025).

---

> > ### Comment · Reviewer_zDhq · 2025-11-26
> >
> > Thank you very much to the authors for their response. I believe the authors’ stated motivation **“the necessity for LLMs to communicate effectively with users whose visual experiences differ significantly from the normal vision assumed by most models”** is highly valuable, and for this reason, I am willing to assign a positive score to this paper. However, I largely agree with reviewer jANV’s viewpoint: despite the paper’s strong motivation, **much of its content reads more like a technical report comparing models trained with different alignment algorithms.** Although the authors have added some experiments during the rebuttal phase and provided a detailed discussion of the system in Appendix C, unfortunately, these additions do not adequately compensate for the core weakness of the current manuscript—**its lack of deep insights and theoretical analysis.** Therefore, I maintain my current score.

---

> ### Author Response · Authors · 2025-12-03
>
> **Lack of deep insights and theoretical analysis.**
> > **Answer:** We thank the reviewer for the response. We address the reviewer's concern of lack of theoretical analysis and deep insights in **section K** of supplementary by grounding our empirical observations into the gradient sturctures of algorithms. To this end, we analyze the underlying gradient structures of GRPO, KTO, and DPO in section K of supplementary. We show:
> >- **GRPO stalls when reward variance is low.** GRPO normalizes rewards within each sampled group, and its update magnitude is proportional to the within-group reward variance. We prove that in cases where reward variance collapses to zero, GRPO's reward-driven gradient is exactly zero, explaining the empirical plateaus observed in our online CLEVR and CUB experiments.
> >- **KTO continues improving even under degenerate rewards.** Unlike GRPO, KTO does not depend on reward heterogeneity. Instead, its gradient is driven by log-likelihood ratios with respect to a fixed reference policy. We show that this gradient remains strictly nonzero even when rewards are constant or uninformative, mirroring the stable improvements KTO achieves across all distortions.
> >- **DPO pushes probability mass toward preferred responses.** For the two-trajectory case, we prove that DPO strictly increases the probability of the preferred response and is maximized in the limit where the model assigns all probability mass to the positive sample. This theoretical behavior aligns with our empirical findings: DPO achieves the strongest peak improvements offline with greedy sampling but may fail under stochastic sampling.
>
> >Taken together, these results provide a unified theoretical grounding for the empirical phenomena observed throughout the main paper: GRPO tends to be conservative but stable because of reward degeneracy leading to vanishing gradients, KTO shows high peak gains because it is robust to reward degeneracy, and DPO inherently encourages probability concentration around preferred trajectories, explaining its strong performance under greedy sampling.

---

### Official Review · Reviewer_jANV · 2025-10-30

**Soundness:** 2
**Presentation:** 2
**Contribution:** 2
**Rating:** 2
**Confidence:** 4

**Summary:**

This paper introduces a variant of the image reference game where a speaker agent with access to undistorted images must communicate with a listener agent who only sees distorted versions. The authors simulate five types of visual distortions inspired by human visual impairments (cataract, AMD, color blindness, tunnel vision, retinal detachment) and evaluate four post-training adaptation methods: SFT and DPO (offline) and KTO and GRPO (online). Experiments on CLEVR and CUB datasets show that these methods can improve communication success, with KTO generally providing the most consistent gains.

**Strengths:**

1. The task of communication under divergent visual perception is practical and relevant, with clear applications to accessibility and multi-agent systems with heterogeneous sensors.
2. The paper thoroughly evaluates multiple methods across two datasets, five distortion types, and includes cross-dataset transfer experiments and tests with different listener models.
3. The paper is well-structured with effective visualizations (particularly Figure 1 and 2) that clearly convey the task setup and qualitative results.

**Weaknesses:**

1. The paper fundamentally presents a benchmarking exercise without extracting meaningful scientific understanding. The core findings (offline methods work with good data, online methods are more stable, adaptation helps with mismatches) are trivial and expected.
2. The paper provides no formal analysis of communication under perceptual constraints, information-theoretic bounds, or theoretical explanation for why certain methods outperform others.
3. The paper is missing for some analytical points that worth to investigate:
- No investigation of what linguistic patterns emerge under different distortions. Although Figure 2 in section 4.3 provides some qualitative study, the explanation is more like a caption of the figure instead of deeper pattern analysis in an quantative way.
- Not a single alignment method consistently outperform others in the analysis and it's confusing about why a certain method performs well under a task setting, and fails in another task setting.
- Missing baseline results like simply prompting the model with the type of visual inconsistency and let it generate the description. Since the zero-shot baseline does not provide a prompt template, we don't know whether this simple baseline has been added or not.
4. The paper simply applies existing methods to a new task without any task-specific innovations or adaptations, which limits its novelty.

**Questions:**

1. Why does KTO outperform other methods? What properties of the algorithm make it suitable for this task?
Could you design task-specific methods that explicitly model the perceptual divergence rather than treating it as a black-box adaptation problem?
2. What linguistic patterns or communication strategies emerge from adaptation? Do models develop generalizable protocols or merely memorize dataset-specific patterns?
3. In Table 2, why the "cataract", "Tunnel Vision", "Detached Retina" and "AMD" columns do not have a bolded number?
4. Directly fine-tuning the models with the preference dataset, but without telling it what kind of perceptual distortions the listeners have can be a confusing choice, the model may simply learn to describe as much as possible instead of describe accordingly for different distortion type.

---

> ### Author Response · Authors · 2025-11-25
>
> We thank the reviewer for the reviews. We address each concern step by step below:
>
> **1. The paper fundamentally presents a benchmarking exercise without extracting meaningful scientific understanding. The core findings are trivial and expected.**
>
> > **Answer:** While our core contribution is a technical investigation of AI-to-AI communication under perceptual mismatch, the **fundamental motivation for modeling divergent visual perception stems from the critical need for accessible and inclusive AI systems**. An estimated 2.2 billion people live with vision impairment or blindness according to the World Health Organization (WHO) [3]. The sheer scale of this population underscores the necessity for LLMs to communicate effectively with users whose visual experiences differ significantly from the ‘normal’ vision assumed by most models. Ensuring that LLMs can adapt their communication to users with divergent perceptual abilities is not merely a matter of engineering efficiency, but a prerequisite for deploying equitable, human-centric AI that serves the entire global population. We introduce a novel setting to investigate these kinds of interactions and provide a benchmarking exercise highlighting various interesting facts through quantitative and qualitative results both. Importantly, our results reveal non-trivial patterns: (i) offline methods produce the strongest peaks; (ii) KTO yields the most reliable improvements across datasets in online settings; (iii) cross-dataset transfer differs sharply between CLEVR→CUB and CUB→CLEVR. Furthermore, we share extensive qualitative analysis in Figure 2 (main manuscript), Figure 4 (supp.), Figure 5 (supp.), Figure 6 (supp.) and Figure 7 (supp.) and linguistic analysis in section J of the supplementary in addition to other useful ablations.**These findings collectively provide new insights into communication under perceptual asymmetry**. Lastly, we will make our code and dataset publicly available to support future research.
>
> **2. The paper provides no formal analysis of communication under perceptual constraints, information-theoretic bounds, or theoretical explanation for why certain methods outperform others.**
>
> > **Answer:** While it is not possible to derive theoretical bounds, we share extensive qualitatives Figure 2 (main manuscript), Figure 4 (supp.), Figure 5 (supp.), Figure 6 (supp.) and Figure 7 (supp.) to shed more light on the workings of these methods. We share the linguistic analysis in section J of the supplementary where we highlight various properties of generations. We look at generations where our model is correct but the base model is incorrect. We compare Flesch score [1] and word count. We observe that the text becomes more readable in the adapted models while overall word count tends to decrease. For example, for detached retina in GRPO, the adapted model has a Flesch score of 55.2 while the base model has a Flesch score of 44.61 and word count for our model is 64 but average word count for adapted model is 538.7. This highlights that the adapted model becomes more concise and understandable. We also observe that KTO Flesch scores are higher than GRPO across most distortions. This shows that KTO aligns the speaker more robustly leading to more concise and readable generations. These results highlight that the language becomes more compact and the alignment process teaches the speaker to effectively communicate with the listener. Lastly, we share the analysis of grayscale distortion  in Figure. 8 of supplementary section J where we observe that the usage of color terms go down after adaptation and other cues (such as background, shape, material etc) are more frequently used which is also confirmed in our qualitative results.

---

> ### Author Response · Authors · 2025-11-25
>
> **3. Missing baseline results like simply prompting the model with the type of visual inconsistency and let it generate the description. Since the zero-shot baseline does not provide a prompt template, we don't know whether this simple baseline has been added or not.**
>
> > **Answer:** We have moved our prompt template from supplementary materials to the method section 3.6 of the main manuscript. We share an ablation over the zero shot performance with more detailed prompts and less detailed prompts in section D table 14 of supplementary. The ablation shows that even with more distortion-aware prompting, models remain significantly below adapted performance confirming that prompting cannot substitute for adaptation. We share the table below.
>
> ### (a) Performance on the **CUB** dataset
>
> | Accuracy Type      | Grayscale | Cataract |
> |--------------------|-----------|----------|
> | **Prompted Acc. [%]** | 74.0      | 80.0     |
> | **Normal Acc. [%]**   | 76.0      | 63.0     |
> | **DPO [%]**   | 85.0     | 89.0     |
>
> ### (b) Performance on the **CLEVR** dataset
>
> | Accuracy Type      | Grayscale | Cataract |
> |--------------------|-----------|----------|
> | **Prompted Acc. [%]** | 50.0      | 56.0     |
> | **Normal Acc. [%]**   | 36.0      | 73.0     |
> | **DPO [%]**   | 75.0      | 77.0    |
>
> Prompted Acc (1st row) shows when we add the distortion information in the prompt and normal (2nd row) shows when we don’t and DPO shows simple DPO finetuning with normal prompt. While we see improved performance in Prompted row (row 1), DPO fine tuning still outperforms significantly showing the need for model-level alignment. Furthermore, we observe lower performance for cataract in CLEVR dataset when we add the distortion information. We share analysis of this case in section D table 15 where we observe that in this case, the model becomes more uncertain and generates repetitive and degenerative sentences leading to drop in performance.

---

> ### Author Response · Authors · 2025-11-26
>
> **4.Why does KTO outperform other methods? What properties of the algorithm make it suitable for this task? Could you design task-specific methods that explicitly model the perceptual divergence rather than treating it as a black-box adaptation problem?**
> > **Answer:**   Recent works have cited GRPO reward function is prone to gradient vanishing in low variance reward datasets [4] which can lead to under exploration. We believe this explains the difference between the two online methods: KTO achieves higher peak performance, while GRPO yields more consistent but conservative improvements. In addition, our linguistic analysis in Section J of the supplementary material shows that KTO achieves higher Flesch readability scores [5], indicating that it produces clearer, more interpretable descriptions. Our goal in this work is to highlight the unique problem of agent communication under divergent visual spaces and to study how different adaptation methods behave in this setting. We also report several interesting empirical findings: KTO is more robust under stochastic sampling (Table 1 and Table 2), offline adaptation remains highly effective even without access to listener dynamics, and SFT demonstrates strong robustness to distribution shifts.
>
>
> **5.In Table 2, why the "cataract", "Tunnel Vision", "Detached Retina" and "AMD" columns do not have a bolded number?**
>
> > **Answer:** We apologize for any mistakes and we have updated the tables.
>
> **6.Directly fine-tuning the models with the preference dataset, but without telling it what kind of perceptual distortions the listeners have can be a confusing choice, the model may simply learn to describe as much as possible instead of describe accordingly for different distortion type.**
>
> > **Answer:**  Our choice to fine-tune the speaker without explicitly revealing the distortion type is intentional and follows the evaluation protocol of Corona et al. (2019) [6]. In this setup, the speaker is deliberately uninformed while the listener sees distorted images. This one-sided communication setting is central to our research question: can a multimodal agent adapt purely through interaction feedback, even without explicit knowledge of the listener’s perceptual impairment?
> Importantly, although the speaker is not told the distortion type, the training setup implicitly conditions the model on the impairment: we train a separate LoRA module for each distortion. Thus, the disability is known implicitly through the dedicated LoRA parameters, which specialize to the corresponding distortion during adaptation. In addition, the preference dataset inherently encodes distortion-aware behavior, positive examples are derived from distorted inputs and negative examples from clean inputs.
> Our results confirm this. Qualitative examples (Fig. 2 in the main paper; Figs. 4–7 in the supplementary) show shifts in descriptions, for example, replacing color cues with shape cues under grayscale, focusing on central regions under tunnel vision, and favoring global or high-contrast cues under cataract. Quantitative analysis in Section J further shows a reduced use of color terms and increased reliance on structural descriptors. These changes demonstrate genuine listener-aligned adaptation rather than over-description.
> In summary, both the dataset design and the fact that each distortion has its own LoRA module ensure that the model does not simply “describe more.” Instead, it learns a distortion-specific communication strategy aligned with what the listener can perceive, which is precisely the emergent behavior we aim to study.

---

> ### Author Response · Authors · 2025-12-03
>
> **7. Not a single alignment method consistently outperforms others in the analysis and it's confusing about why a certain method performs well under a task setting, and fails in another task setting.**
>
> > **Answer:** In the main manuscript (tables 1 and 2), we report the average results across 10 runs with stochastic sampling (temperature = 0.7). Under this setting, KTO yields substantial overall gains of +7.62% on CLEVR for test-pick accuracy (best among online methods). This is the second-highest overall, followed by DPO (test-pick accuracy gain of +8.32%). Still, KTO outperforms DPO across 3/5 distortions, and DPO's improvement is driven by high peak gains in detached retina and a lower drop in performance for AMD. In CUB, KTO shows +5.80% which is the best overall. However, it is beaten by GRPO in 3/5 distortions (AMD, Detached Retina, and Tunnel Vision) in test-pick performance while KTO outperforms GRPO in 3/5 distortions in CLEVR dataset (main table 1).  KTO also consistently outperforms other methods on val-pick performance under stochastic sampling, demonstrating reliability in the absence of known task characteristics.
> We share the greedy sampling results in Supplementary Section A, Tables 8, 9, and 10. These results consistently show that DPO outperforms all other methods in test-pick accuracy. This suggests that, if the disability is known, offline methods should be used (DPO wins at least 3/5 across all datasets in Sup. tables 8-19).
> However, if the disability is unknown or sample diversity is desired (stochastic sampling), the choice of algorithm depends on the task characteristics, since no single method outperforms others across all or most distortions and datasets. To this end, we share a comprehensive study on five different distortions. Lastly, we share a protocol for developing a preference dataset for robust offline adaptation, along with a demonstration of how online adaptation can be achieved.

---

> ### Author Response · Authors · 2025-12-03
>
> **8.Paper is lacking meaningful scientific findings**
>
> > **Answer:** We extend our initial response to this question by providing theoretical analysis in **section K** of supplementary by grounding our empirical observations into the gradient sturctures of algorithms. To this end, we analyze the underlying gradient structures of GRPO, KTO, and DPO in section K of supplementary. We show:
> >- **GRPO stalls when reward variance is low.** GRPO normalizes rewards within each sampled group, and its update magnitude is proportional to the within-group reward variance. We prove that in cases where reward variance collapses to zero, GRPO's reward-driven gradient is exactly zero, explaining the empirical plateaus observed in our online CLEVR and CUB experiments.
> >- **KTO continues improving even under degenerate rewards.** Unlike GRPO, KTO does not depend on reward heterogeneity. Instead, its gradient is driven by log-likelihood ratios with respect to a fixed reference policy. We show that this gradient remains strictly nonzero even when rewards are constant or uninformative, mirroring the stable improvements KTO achieves across all distortions.
> >- **DPO pushes probability mass toward preferred responses.** For the two-trajectory case, we prove that DPO strictly increases the probability of the preferred response and is maximized in the limit where the model assigns all probability mass to the positive sample. This theoretical behavior aligns with our empirical findings: DPO achieves the strongest peak improvements offline with greedy sampling but may fail under stochastic sampling.
>
> >Taken together, these results provide a unified theoretical grounding for the empirical phenomena observed throughout the main paper: GRPO tends to be conservative but stable because of reward degeneracy leading to vanishing gradients, KTO shows high peak gains because it is robust to reward degeneracy, and DPO inherently encourages probability concentration around preferred trajectories, explaining its strong performance under greedy sampling.
>
>
> [1] Klare, George R. "Assessing readability." Reading research quarterly (1974): 62-102.
>
> [2] Nwabueze, Josephine Nneka. "BARRIERS GLOBALLY FACED BY PERSNS WITH DISABILITIES." Special needs education from the lens of interdisciplinary dialogue: a festschrift in honour of prof. Emeka d. Ozoji 3 (2023).
>
> [3] World Health Organization and The Lancet Global Health Commission on Global Eye Health. World Report on Vision. World Health Organization, 2021. URL https://www.who.int/publications/i/ item/9789241517402.
>
> [4] Leng, Sicong, et al. "MMR1: Enhancing Multimodal Reasoning with Variance-Aware Sampling and Open Resources." arXiv preprint arXiv:2509.21268 (2025).
>
> [5] Klare, George R. "Assessing readability." Reading research quarterly (1974): 62-102.
>
> [6] Corona, Rodolfo, Stephan Alaniz, and Zeynep Akata. "Modeling conceptual understanding in image reference games." arXiv preprint arXiv:1910.04872 (2019).

---

### Official Review · Reviewer_bxcA · 2025-11-01

**Soundness:** 3
**Presentation:** 3
**Contribution:** 3
**Rating:** 6
**Confidence:** 4

**Summary:**

This paper introduces a new setting called “divergent visual spaces”, where two multimodal agents see the same world differently. One observes the clean image, while the other receives a distorted version (simulating human vision impairments).
Using image reference games, the authors analyze how such mismatches affect communication and compare offline alignment (SFT, DPO) versus online alignment (KTO, GRPO) strategies.

**Strengths:**

- Novel problem formulation: Studies communication under perceptual asymmetry, a realistic and underexplored challenge for multimodal agents, extending beyond identical sensory inputs.
- Well-designed benchmark: Provides new preference dataset (distorted vs undistorted prompts) and controlled visual impairments inspired by human conditions.
- Strong qualitative insight: Demonstrates how models adapt their descriptions to match limited visual cues (e.g., from color-based to structure-based features).

**Weaknesses:**

1. On Dataset Curation and Sample Construction
- How the paired image samples (target vs. distractor) were selected? My understanding is that they should be semantically similar but not identical to make the reference game non-trivial. However, this similarity control can be hard to standardize.
Could the authors clarify the criteria or process used for curating these pairs and ensuring balanced difficulty across distortions?
2. On the Generalization of Training Effects
- Beyond the reference game, does this training improve the model’s general image understanding or fine-grained perception?
If the speaker learns to “see through distortions,” could this act as a form of RL-based perceptual sharpening, transferable to other downstream tasks (e.g., captioning, general image understanding)?
- Conceptually, the training objective appears to mix generative (speaker) and discriminative (listener) alignment. So it's better to evaluate the model on image understanding tasks like MMBench, MMVet, etc.
3. On Evaluation Robustness (False Positives / False Negatives)
- The current evaluation may suffer from ambiguity bias: a coarse or underspecified description could accidentally fit both candidate images, leading to inflated listener accuracy (false positives).
- Conversely, fine-grained but valid descriptions might mismatch due to listener limitations (false negatives).
- Have the authors considered prompt variants (e.g., “Describe this image in detail” vs. short prompts) to quantify this sensitivity? Would richer prompts yield more reliable alignment and higher communication success?

**Questions:**

See Weakness

---

> ### Author Response · Authors · 2025-11-25
>
> We thank the reviewer for the thoughtful reviews. We address each concern step by step below:
>
> **1. How the paired image samples (target vs. distractor) were selected? My understanding is that they should be semantically similar but not identical to make the reference game non-trivial. However, this similarity control can be hard to standardize. Could the authors clarify the criteria or process used for curating these pairs and ensuring balanced difficulty across distortions?**
>
> > **Answer:** The difficulty comes from the agent's limitations (e.g., color blindness), not just the image similarity. Random images from the same dataset often share shapes/materials (CLEVR), so if an attribute is removed, they become harder to distinguish. The same is true for the CUB dataset which is a finegrained dataset where even birds of different classes are difficult to distinguish by construction. This difficulty is further amplified by the introduced distortions. Section 3.4 explains that we do not perform semantic filtering of distractors; instead, distortions themselves modulate difficulty, consistent with our goal of studying communication under perceptual asymmetry rather than dataset-induced similarity constraints.
>
> **2. Beyond the reference game, does this training improve the model’s general image understanding or fine-grained perception?**
>
> > **Answer:** We show generalization capability in Table 3,4 of the main manuscript that shows cross dataset generalization demonstrating that they learn transferable techniques instead of memorizing one dataset. Concretely, training on CLEVR and testing on CUB yields up to +14.1% (SFT, Grayscale), showing transfer of distortion-robust communication. The reverse direction yields smaller gains due to CLEVR’s compositional nature, but still improves up to +10.7% (GRPO, Tunnel Vision). We conclude that optimizing on one dataset improves model capability on a more general data setting. However, we do not necessarily expect these task-specific improvements to translate to other general model capabilities as we tackle a fundamentally different problem that has not been considered in other benchmarks.
>
> **3. Conversely, fine-grained but valid descriptions might mismatch due to listener limitations (false negatives).**
>
> > **Answer:** We share many qualitatives in Figure 2 (main manuscript), Figure 4 (supp.), Figure 5 (supp.), Figure 6 (supp.) and Figure 7 (supp.) to show that the success is coming from faithful alignment with the listener. We have added a linguistic analysis in section J of supplementary to further study this. Our linguistic analysis confirms that adapted models avoid inaccessible fine-grained cues: adapted models reduce word count and improve readability (higher Flesch scores) compared to base models, which often produce long, repetitive sentences misaligned with distorted perception.
>
> **3b. Have the authors considered prompt variants (e.g., “Describe this image in detail” vs. short prompts) to quantify this sensitivity? Would richer prompts yield more reliable alignment and higher communication success?**
>
> > **Answer:** We share an ablation over the zero shot performance with more detailed prompts and less detailed prompts in section D table 14 of supplementary. Furthermore, we have also moved our prompt template from supplementary materials to the method section 3.6 of the main manuscript. The ablation shows that prompt detail level affects ZS performance but does not close the gap to adapted models, adaptation still yields significantly higher gains, confirming the necessity of model-level alignment. We share results in the following table.
> ### (a) Performance on the **CUB** dataset
>
> | Accuracy Type      | Grayscale | Cataract |
> |--------------------|-----------|----------|
> | **Prompted Acc. [%]** | 74.0      | 80.0     |
> | **Normal Acc. [%]**   | 76.0      | 63.0     |
> | **DPO [%]**   | 85.0     | 89.0     |
>
> ### (b) Performance on the **CLEVR** dataset
>
> | Accuracy Type      | Grayscale | Cataract |
> |--------------------|-----------|----------|
> | **Prompted Acc. [%]** | 50.0      | 56.0     |
> | **Normal Acc. [%]**   | 36.0      | 73.0     |
> | **DPO [%]**   | 75.0      | 77.0    |
>
> Prompted Acc (1st row) shows when we add the distortion information in the prompt and normal (2nd row) shows when we don’t and DPO shows simple DPO finetuning with normal prompt. While we see improved performance in Prompted row (row 1), DPO fine tuning still outperforms significantly showing the need for model-level alignment. Furthermore, we observe lower performance for cataract in CLEVR dataset when we add the distortion information. We share analysis of this case in section D table 15 where we observe that in this case, the model becomes more uncertain and generates repetitive and degenerative sentences leading to drop in performance.

---

### Official Review · Reviewer_vz4H · 2025-11-03

**Soundness:** 2
**Presentation:** 3
**Contribution:** 2
**Rating:** 2
**Confidence:** 4

**Summary:**

The paper studies how multimodal agents communicate when they perceive images differently. Using an image reference game, one agent sees clear images while another sees distorted ones mimicking human visual impairments like cataract or color blindness. The authors test four adaptation methods: offline and online. Offline methods use preference data contrasting distortion-aware vs. normal descriptions; online ones learn from feedback during interaction. Experiments on CLEVR and CUB datasets show all methods improve over baselines. DPO performs best when high-quality data exist, while KTO gives the most consistent results across distortions. Adapted speakers learn to emphasize features the listener can still perceive. Cross-dataset and cross-model tests show offline methods give larger but less stable gains, whereas online methods are more reliable. Overall, offline adaptation works best when supervision is available; online adaptation is better for dynamic, unknown perceptual gaps.

**Strengths:**

1. While most visual-language alignment studies focus on human-AI interaction, this paper provides valuable insight by considering multi-agent systems instead.

2. The qualitative analysis results were particularly impressive.

**Weaknesses:**

1. Although the paper emphasizes agent systems and interactive settings as its main focus, there is almost no analysis of multi-turn interactions, nor any investigation of essential agent capabilities such as tool use.

2. Relatedly, the paper lacks elements that support the generalizability of its findings such as diversity in benchmarks and models. It appears that only Qwen2.5-VL-32B was used; including results from other VLMs or model sizes would strengthen the claims. This was the most decisive factor in my final rating, and demonstrating broader generalizability would significantly improve the paper.

3. The overall message of the research is not clearly conveyed. It’s unclear whether online methods are definitively better than offline ones, or if each has advantages in specific situations. The paper does not clearly specify when each method performs better. Moreover, differing trends across benchmarks make the results difficult to interpret.

**Questions:**

1. The paper emphasizes multi-agent communication, yet most experiments involve single-turn interactions. How would the proposed methods scale to multi-turn or continuous dialogue settings between agents?

2. Have the authors considered incorporating tool use or memory capabilities that are crucial for realistic agent systems into their framework? How might divergent visual spaces affect such abilities?

3. The study relies on simulated perceptual distortions. How closely do these distortions correspond to real-world sensor or perception mismatches in practical multi-agent systems?

4. The experiments use Qwen2.5-VL-32B (and partly InternVL3.5). Would the conclusions hold across different model architectures or smaller/larger model sizes?

5. How sensitive are the results to the choice of visual-language model? For instance, would models like GPT-4V or LLaVA show similar adaptation behaviors?

6. Could the authors test on additional benchmarks or more naturalistic visual environments to demonstrate broader generalizability beyond CLEVR and CUB?

7. Are there any ablation studies showing how much each distortion type (e.g., color loss vs. occlusion) affects alignment performance?

8. The paper’s main message about when to use offline versus online adaptation seems unclear. Can the authors provide a clearer guideline or decision boundary for practitioners?

9. Some results show inconsistent trends across benchmarks does this suggest that the effectiveness of each method depends strongly on task characteristics?

10. How do the authors define or measure "alignment" between agents purely through task success rates, or is there a measure of interpretability or communicative grounding?

---

> ### Author Response · Authors · 2025-11-25
>
> We thank the reviewer for the thoughtful reviews. We answer each concern one by one below.
>
> **1. The paper emphasizes multi-agent communication, yet most experiments involve single-turn interactions. How would the proposed methods scale to multi-turn or continuous dialogue settings between agents?**
>
> > **Answer:** To start from a controlled setting, we adopt the evaluation protocol from Corona et al. [6] and extend it to free form text generation starting from an uninformed speaker agent. This protocol restricts the communication to be one sided. In this scenario we are specifically studying how the agent adapts to the listener when there is no information. To extend it to continuous dialogue settings between agents, we would let the speaker and listener interact until termination or a specific amount of interactions have been achieved. The interactions leading to success will be positive trajectories and others negative.
>
> **2. Have the authors considered incorporating tool use or memory capabilities that are crucial for realistic agent systems into their framework? How might divergent visual spaces affect such abilities?**
>
> > **Answer:** We argue that the image reference games we used serve as a ‘test bed’ for evaluating agents under divergent visual spaces by establishing a controlled setting. This facilitates further research into memory capabilities and tool use. At the same time, our method of training LoRA adapters has similarities to memory methods, as the model is able to incorporate knowledge about the listener’s impairments permanently into their weights. Since the LoRA is individual for each listener, it can be adaptively swapped depending on the conversation partner. Further studies on how divergent visual spaces affect the model’s memory and tooling capabilities are an interesting future research direction. We also note that our results show consistent improvements even without explicit memory e.g., online KTO achieves up to +17.7% (CUB, AMD) and +36.4% (CLEVR, Grayscale) improvement, indicating that the core alignment challenge can be addressed even without persistent memory.
>
> **3. The study relies on simulated perceptual distortions. How closely do these distortions correspond to real-world sensor or perception mismatches in practical multi-agent systems?**
>
> > **Answer:** The perceptual distortions are derived from real human eye diseases and closely follow their definitions [4]. We explain the challenges of our distortions in main manuscript section 3.2. However, these distortions are applicable to camera lenses as well as they introduce occlusion or reduce available information (perceptual noise), distortions that have been studied well in the computer vision domain [1,2,3]. As detailed in Section 3.2, each distortion is constructed directly from biomedical descriptions: cataract is modeled with blurring + contrast loss + random clouding artifacts; AMD masks the central field; tunnel vision removes peripheral vision; detached retina adds irregular occlusions; and grayscale simulates color-channel loss. These reflect real optical degradation patterns rather than simple noise [4].

---

> ### Author Response · Authors · 2025-11-25
>
> **4. The experiments use Qwen2.5-VL-32B (and partly InternVL3.5). Would the conclusions hold across different model architectures or smaller/larger model sizes?**
>
> > **Answer:**
> > *   We apologize for this misunderstanding. We use Qwen 2.5-VL 32B only for annotating samples for offline adaptation. We use Qwen 2.5-VL 7B for training. To extend our analysis to models of different sizes, we conducted experiments combining Qwen2.5-VL 3B and 7B in different speaker-listener configurations trained with GRPO and KTO. We do this experiment with 4 distortions, Age related macular degeneration, cataract, detached retina and grayscale on the CUB dataset. We report results in Section B of supplementary and table 11 with detailed discussion and in the following table. Concretely, these heterogeneous-size experiments show that while absolute accuracy varies, pairing a smaller, computationally cheaper speaker model with a larger, more capable listener model provides a strong initial baseline that can be further enhanced through adaptation to achieve.
>
> | Method | Speaker            | Listener           | Type | AMD | Cataract | Detached Retina | Grayscale | Average |
> |--------|---------------------|---------------------|------|-----|----------|------------------|-----------|----------|
> | **GRPO** | Qwen-2.5 VL 3B     | Qwen-2.5 VL 3B      | Base | 89.0 | 67.0 | 98.0 | 94.0 | 87.0 |
> |        |                     |                     | Max  | 91.0 (+2.0) | 67.0 (+0.0) | 99.0 (+1.0) | 95.0 (+1.0) | 88.0 (+1.0) |
> |        | Qwen-2.5 VL 3B     | Qwen-2.5 VL 7B      | Base | 94.0 | 79.0 | 98.0 | 84.0 | 88.8 |
> |        |                     |                     | Max  | 98.0 (+4.0) | 86.0 (+7.0) | 100.0 (+2.0) | 84.0 (+0.0) | 92.0 (+3.2) |
> |        | Qwen-2.5 VL 7B     | Qwen-2.5 VL 3B      | Base | 92.0 | 64.0 | 97.0 | 93.0 | 86.5 |
> |        |                     |                     | Max  | 97.0 (+5.0) | 72.0 (+8.0) | 98.0 (+1.0) | 98.0 (+5.0) | 91.3 (+4.8) |
> |        | Qwen-2.5 VL 7B     | Qwen-2.5 VL 7B      | Base | 90.0 | 63.0 | 76.0 | 88.0 | 81.4 |
> |        |                     |                     | Max  | 93.0 (+3.0) | 82.0 (+19.0) | 77.0 (+1.0) | 93.0 (+5.0) | 87.6 (+6.2) |
> | **KTO** | Qwen-2.5 VL 3B     | Qwen-2.5 VL 3B      | Base | 88.0 | 67.0 | 96.0 | 94.0 | 86.3 |
> |        |                     |                     | Max  | 89.0 (+1.0) | 70.0 (+3.0) | 100.0 (+4.0) | 98.0 (+4.0) | 89.5 (+3.2) |
> |        | Qwen-2.5 VL 3B     | Qwen-2.5 VL 7B      | Base | 94.0 | 80.0 | 98.0 | 80.0 | 88.0 |
> |        |                     |                     | Max  | 96.0 (+2.0) | 85.0 (+5.0) | 99.0 (+1.0) | 82.0 (+2.0) | 90.5 (+2.5) |
> |        | Qwen-2.5 VL 7B     | Qwen-2.5 VL 3B      | Base | 92.0 | 67.0 | 96.0 | 93.0 | 87.0 |
> |        |                     |                     | Max  | 93.0 (+1.0) | 75.0 (+8.0) | 97.0 (+1.0) | 97.0 (+4.0) | 90.8 (+3.8) |
> |        | Qwen-2.5 VL 7B     | Qwen-2.5 VL 7B      | Base | 90.0 | 63.0 | 76.0 | 88.0 | 81.4 |
> |        |                     |                     | Max  | 94.0 (+4.0) | 81.0 (+18.0) | 79.0 (+3.0) | 93.0 (+5.0) | 87.8 (+6.4) |

---

> > ### Author Response · Authors · 2025-11-25
> >
> > **5. How sensitive are the results to the choice of visual-language model? For instance, would models like GPT-4V or LLaVA show similar adaptation behaviors?**
> >
> > > **Answer:** Based on our heterogeneous model ablation in section 5.2, we can argue that choice of models can affect the results significantly. For example, when switching from a Qwen listener to an InternVL3.5 listener, improvements shrink for already strong distortions (near-ceiling accuracy on Tunnel Vision and Detached Retina) while remaining large for weaker listeners (CLEVR Grayscale improves by +34.4% with KTO). This indicates that adaptation benefits depend strongly on the listener’s baseline robustness.
> >
> > **6. Could the authors test on additional benchmarks or more naturalistic visual environments to demonstrate broader generalizability beyond CLEVR and CUB?**
> >
> > > **Answer:** Thanks for pointing this out. We used imagenet benchmark for a more naturalistic setting. We trained Qwen 7B as a listener and a speaker both. Below, we share the maximum test accuracy achieved. We have added these results in section A table 8 Supplementary. Concretely, on ImageNet, adaptation improves accuracy from 83.0% (ZS) to as high as 88.0% (+5.0) with DPO and +4.4 average gain with GRPO, demonstrating strong generalization to natural images.
> >
> >
> > | Method | AMD                 | Cataract               | Grayscale               | Tunnel Vision            | Detached Retina          | Avg.                 |
> > |--------|----------------------|-------------------------|--------------------------|---------------------------|---------------------------|------------------------|
> > | **ZS** | 89.0                | 75.0                   | 79.0                    | 85.0                     | 87.0                     | 83.0                 |
> > | **KTO** | 92.0 / 92.0 (+3.0 / +3.0) | 82.0 / 74.0 (+7.0 / -1.0) | 75.0 / 75.0 (-4.0 / -4.0) | 89.0 / 89.0 (+4.0 / +4.0) | 90.0 / 90.0 (+3.0 / +3.0) | 85.6 / 84.0 (+2.6 / +1.0) |
> > | **GRPO** | **95.0 / 95.0 (+6.0 / +6.0)** | 79.0 / 78.0 (+4.0 / +3.0) | 83.0 / 83.0 (+4.0 / +4.0) | 91.0 / 91.0 (+6.0 / +6.0) | 90.0 / 90.0 (+3.0 / +3.0) | 87.6 / 87.4 (+4.6 / +4.4) |
> > | **DPO** | 94.0 / 94.0 (+5.0 / +5.0) | 82.0 / 76.0 (+7.0 / +1.0) | **84.0 / 81.0 (+5.0 / +2.0)** | **92.0 / 90.0 (+7.0 / +5.0)** | 88.0 / 88.0 (+1.0 / +1.0) | **88.0 / 85.8 (+5.0 / +2.8)** |
> > | **SFT** | **95.0 / 95.0 (+6.0 / +6.0)** | **84.0 / 75.0 (+9.0 / +0.0)** | 79.0 / 79.0 (+0.0 / +0.0) | 85.0 / 85.0 (+0.0 / +0.0) | **92.0 / 92.0 (+5.0 / +5.0)** | 87.0 / 85.2 (+4.0 / +2.2) |
> >
> >
> >
> > **7. Are there any ablation studies showing how much each distortion type (e.g., color loss vs. occlusion) affects alignment performance?**
> >
> > > **Answer:** In our setting, color loss and occlusion represent two broad categories of visual information degradation. When we reinterpret our results through this lens, we observe that degradations involving severe color loss lead to the largest performance improvements after adaptation (e.g., gains of up to +36.4%). Degradations dominated by occlusion show moderate but consistent improvements (typically +3–7%).

---

> > ### Comment · Reviewer_vz4H · 2025-11-26
> >
> > Thank you for your comments! All of my concerns and questions have been addressed, I’ll raise my score to 6.

---

> ### Author Response · Authors · 2025-11-25
>
> **8. Some results show inconsistent trends across benchmarks. Does this suggest that the effectiveness of each method depends strongly on task characteristics? Can the authors provide a clearer guideline or decision boundary for practitioners?**
>
> > **Answer:** In the main manuscript (tables 1 and 2), we report the average results across 10 runs with stochastic sampling (temperature = 0.7). Under this setting, KTO yields substantial overall gains of +7.62% on CLEVR for test-pick accuracy (best among online methods). This is the second-highest overall, followed by DPO (test-pick accuracy gain of +8.32%). Still, KTO outperforms DPO across 3/5 distortions, and DPO's improvement is driven by high peak gains in detached retina and a lower drop in performance for AMD. In CUB, KTO shows +5.80% which is the best overall. However, it is beaten by GRPO in 3/5 distortions (AMD, Detached Retina, and Tunnel Vision) in test-pick performance while KTO outperforms GRPO in 3/5 distortions in CLEVR dataset (main table 1).  KTO also consistently outperforms other methods on val-pick performance under stochastic sampling, demonstrating reliability in the absence of known task characteristics.
> We share the greedy sampling results in Supplementary Section A, Tables 8, 9, and 10. These results consistently show that DPO outperforms all other methods in test-pick accuracy. This suggests that, if the disability is known, offline methods should be used (DPO wins at least 3/5 across all datasets in Sup. tables 8-19).
> However, if the disability is unknown or sample diversity is desired (stochastic sampling), the choice of algorithm depends on the task characteristics, since no single method outperforms others across all or most distortions and datasets. To this end, we share a comprehensive study on five different distortions. Lastly, we share a protocol for developing a preference dataset for robust offline adaptation, along with a demonstration of how online adaptation can be achieved.
>
> **10. How do the authors define or measure "alignment" between agents purely through task success rates, or is there a measure of interpretability or communicative grounding?**
>
> > **Answer:** We define alignment as task success. If the model succeeds in improving the performance, it is more aligned than the model that does not succeed. Furthermore, we confirm from many qualitative results in Figure 2 (main manuscript), Figure 4 (supp.), Figure 5 (supp.), Figure 6 (supplementary) and Figure 7 (supp.) that indeed, the success comes from the speaker aligning its descriptions with the listener's perspective. In addition, Section J linguistic analysis shows adapted models produce more readable text (higher Flesch scores, fewer repetitions) and shorter, more concise descriptions, indicating improved communicative grounding beyond accuracy. For example, for detached retina in GRPO, the adapted model has a Flesch score of 55.2 while the base model has a Flesch score of 44.61 and word count for our model is 64 but average word count for adapted model is 538.7. This highlights that the adapted model becomes more concise and understandable. We also observe that KTO Flesch scores are higher than GRPO across most distortions.
>
>
> [1] Ke, Lei, Yu-Wing Tai, and Chi-Keung Tang. "Deep occlusion-aware instance segmentation with overlapping bilayers." Proceedings of the IEEE/CVF conference on computer vision and pattern recognition. 2021.
>
> [2] Chen, Peixian, et al. "Occlude them all: Occlusion-aware attention network for occluded person re-id." Proceedings of the IEEE/CVF international conference on computer vision. 2021.
>
> [3] Liang, Jingyun, et al. "Swinir: Image restoration using swin transformer." Proceedings of the IEEE/CVF international conference on computer vision. 2021.
>
> [4] Wikimedia Commons. (2013, October 4). Eye disease simulation. Retrieved from https://commons.wikimedia.org/wiki/Eye_disease_simulation
>
> [6] Corona, Rodolfo, Stephan Alaniz, and Zeynep Akata. "Modeling conceptual understanding in image reference games." arXiv preprint arXiv:1910.04872 (2019).

---

### Author Response · Authors · 2025-11-25
**Global Message for the reviewers**

We thank the reviewers for their thoughtful feedback. Reviewers **vz4H** and **zDhq** both highlight the value and social relevance of our insights, noting our strong execution and systematic experimental design. Reviewers **bxcA** and **jANV** appreciate the novelty of our problem formulation, the well-designed benchmark, and the thorough structure and qualitative strength of our evaluation. We mention here the common concerns and the rest in the individual responses.
We acknowledge that reviewers **vz4H** and **jANV** find the results ambiguous. In the main manuscript (tables 1 and 2), we report the average results across 10 runs with stochastic sampling (temperature = 0.7). Under this setting, KTO yields substantial overall gains of +7.62% on CLEVR for test-pick accuracy (best among online methods). This is the second-highest overall, followed by DPO (test-pick accuracy gain of +8.32%). Still, KTO outperforms DPO across 3/5 distortions, and DPO's improvement is driven by high peak gains in detached retina and a lower drop in performance for AMD. In CUB, KTO shows +5.80% which is the best overall. However, it is beaten by GRPO in 3/5 distortions (AMD, Detached Retina, and Tunnel Vision) in test-pick performance while KTO outperforms GRPO in 3/5 distortions in CLEVR dataset (main table 1).  KTO also consistently outperforms other methods on val-pick performance under stochastic sampling, demonstrating reliability in the absence of known task characteristics.

We share the greedy sampling results in Supplementary Section A, Tables 8, 9, and 10. These results consistently show that DPO outperforms all other methods in test-pick accuracy. This suggests that, if the disability is known, offline methods should be used (DPO wins at least 3/5 across all datasets in Sup. tables 8-19).

However, if the disability is unknown or sample diversity is desired (stochastic sampling), the choice of algorithm depends on the task characteristics, since no single method outperforms others across all or most distortions and datasets. To this end, we share a comprehensive study on five different distortions. Lastly, we share a protocol for developing a preference dataset for robust offline adaptation, along with a demonstration of how online adaptation can be achieved.

We also acknowledge that reviewers **jANV** and **zDhq** find our work lacking novelty. While our core contribution is a technical investigation of AI-to-AI communication under perceptual mismatch, the fundamental motivation for modeling divergent visual perception stems from the critical need for accessible and inclusive AI systems. Visual impairments are a widespread global health issue, with an estimated 2.2 billion people living with vision impairment or blindness according to the World Health Organization [1]. The sheer scale of this population underscores the necessity for LLMs to communicate effectively with users whose visual experiences differ significantly from the ``normal'' vision assumed by most models. Ensuring that LLMs can adapt their communication to users with divergent perceptual abilities is not merely a matter of engineering efficiency, but a prerequisite for deploying equitable, human-centric AI that serves the entire global population.

[1]  World Health Organization and The Lancet Global Health Commission on Global Eye Health. World Report on Vision. World Health Organization, 2021. URL https://www.who.int/publications/i/ item/9789241517402.

---

### Meta-Review · Area_Chair_yfdR · 2026-01-06

**Summary:**

The paper studies an interesting and novel setting of agentic systems, in which two agents with divergent perceptual inputs need to communicate. The authors formulate the problem as an image reference game and investigate the effectiveness of different post-training algorithms under varying degrees of visual degradation. The reviewers find the story appealing and consider the multi-agent setup with heterogeneous sensory inputs to be novel. However, they are also concerned about the experimental designs: testing only on CLEVR/CUB,  not enough technical depth (ie, simply running existing algorithms without providing much insights), experimented only with limited VLMs, etc. During the discussion period, the authors provided additional experiments requested by the reviewers. At the end, one negative raise their score from reject to borderline accept, while the rest remains their original evaluation. While the paper has merit, the AC agrees that many aspects could have been improved and explored in greater depth. The authors took a number of actions during the rebuttal period; however, the necessary changes would likely require a major revision. Importantly, no reviewer is willing to champion the paper. After extensive discussion, the AC decides to reject the paper. The authors are encouraged to incorporate the feedbacks from the reviewers and submit to a future venue.

**Reviewer Concerns:**

The authors provide additional empirical experiments on a few new datasets and models; however, the paper still reads as a relatively straightforward adaptation of existing algorithms to a new setting, without substantial insights.

**Reviewer Scores:**

As mentioned above, while the authors took a number of actions during the rebuttal period, the necessary changes would likely require a major revision. One reviewer raised their score from negative to borderline accept, while the others remained unchanged. That said, no reviewer was willing to champion the paper.

---

### Decision · Program_Chairs · 2026-01-26

Reject